# SWI/SNF and the histone chaperone Rtt106 drive expression of the Pleiotropic Drug Resistance network genes

Vladislav N. Nikolov[1], Dhara Malavia[1,2] & Takashi Kubota [1✉]

The Pleiotropic Drug Resistance (PDR) network is central to the drug response in fungi, and its overactivation is associated with drug resistance. However, gene regulation of the PDR network is not well understood. Here, we show that the histone chaperone Rtt106 and the chromatin remodeller SWI/SNF control expression of the PDR network genes and confer drug resistance. In *Saccharomyces cerevisiae*, Rtt106 specifically localises to PDR network gene promoters dependent on transcription factor Pdr3, but not Pdr1, and is essential for Pdr3-mediated basal expression of the PDR network genes, while SWI/SNF is essential for both basal and drug-induced expression. Also in the pathogenic fungus *Candida glabrata*, Rtt106 and SWI/SNF regulate drug-induced PDR gene expression. Consistently, loss of Rtt106 or SWI/SNF sensitises drug-resistant *S. cerevisiae* mutants and *C. glabrata* to antifungal drugs. Since they cooperatively drive PDR network gene expression, Rtt106 and SWI/SNF represent potential therapeutic targets to combat antifungal resistance.

[1] Institute of Medical Sciences, School of Medicine, Medical Sciences & Nutrition, University of Aberdeen, Foresterhill, Aberdeen AB25 2ZD, UK. [2] Present address: MRC Centre for Medical Mycology, Geoffrey Pope Building, University of Exeter, Stocker Road, Exeter EX4 4QD, UK. ✉email: t.kubota@abdn.ac.uk

The global incidence of life-threatening *Candida* infections stands at around 400,000 cases annually, with the emergence of multidrug-resistant strains of *Candida* species being a major global health concern[1–3]. Undermining the utility of antifungal drugs, the Pleiotropic Drug Resistance (PDR) network of fungi works as the first line of defence against a broad range of xenobiotics[4,5], and is central to the fungal drug response[6,7]. Among the PDR network gene products, Pdr5, a major ATP-binding cassette (ABC) multidrug transporter, is critical for drug resistance in *Saccharomyces cerevisiae*[8], a fungus closely related to the pathogenic fungus *Candida glabrata*. Pdr5 pumps a variety of drugs out of cells at the expense of ATP hydrolysis, and Pdr5 overexpression is one of the main causes of drug resistance in *S. cerevisiae*[8]. Drug resistance to azole antifungals in clinical isolates of *C. glabrata* is almost exclusively associated with overexpression of multidrug transporter genes *CgCDR1* (homolog of *ScPDR5*) and *CgCDR2*[9].

The expression of *PDR5* and the other PDR network genes in *S. cerevisiae* is regulated by Pdr1 and Pdr3, key transcription factors whose functions are partially redundant[10–12]. Pdr1 and Pdr3 form a heterodimer or homodimers, which through their Zn(2)-Cys(6) finger DNA-binding domains[13] recognise the PDR responsive element (PDRE: 5′-TCCG(C/T)GGA-3′) present in promoters of the PDR network genes. Pdr1 and Pdr3 together sustain basal gene expression, and Pdr1 activates increased gene expression in response to direct binding to the azole antifungal ketoconazole[7]. While Pdr3 also can directly bind ketoconazole[7], it has not been shown whether its activity is stimulated by ketoconazole binding. Hyperactivation of Pdr1 or Pdr3 is the main cause of overexpression of *PDR5* leading to drug resistance. Such hyperactivation can be caused by gain-of-function (GOF) mutations in *PDR1* or *PDR3*[14,15] or mitochondrial dysfunction (e.g. loss of the mitochondrial genome), the latter of which only stimulates Pdr3-dependent gene expression[16]. In *C. glabrata* also, overexpression of the drug transporter genes is caused by GOF mutations in the transcription factor Pdr1[9]. Given the importance of multidrug transporter levels for drug resistance, it is vital to understand the regulation of their expression.

Accessibility of gene promoters is a key factor in gene regulation[17,18]. Recruitment of transcription machinery to promoters is restricted by nucleosomes[19]. Histone chaperones and ATP-dependent chromatin-remodelling complexes are accordingly important for gene regulation given they regulate nucleosome dynamics at promoters, restricting or increasing promoter accessibility for recruitment of transcription factors and RNA polymerase II (Pol II)[19–23].

Rtt106, a fungal-specific histone chaperone, functions in histone deposition during DNA replication and transcription elongation in *S. cerevisiae*, and preferentially binds H3K56-acetylated histone H3-H4[24–26]. Loss of Rtt106 causes aberrant 'cryptic' transcription initiation within gene bodies[26] and compromised silencing at the mating loci and telomeres[27,28], likely due to defective nucleosome assembly[26,27,29].

Apart from its role in histone deposition during replication and transcription elongation, Rtt106 regulates the activity of the histone gene promoters during the cell cycle in *S. cerevisiae*[30]. Rtt106 localises at the promoters of three out of four histone gene pairs (*HTA1-HTB1*, *HHT1-HHF1* and *HHT2-HHF2*, but not *HTA2-HTB2*), and restricts their transcription to S phase, when new histone supply is required to package newly synthesised DNA[31,32]. Outside S phase, Rtt106 forms a repressive complex together with other histone chaperones HIR and Asf1 to recruit RSC (Remodel the Structure of Chromatin), an essential ATP-dependent chromatin-remodelling complex, to the histone promoters[31–33]. In contrast, in the late G1 and early S phase Rtt106 recruits SWI/SNF, another ATP-dependent chromatin-remodelling complex, which activates transcription of the histone genes by releasing HIR- and RSC-mediated repression[33,34]. Rtt106 recruitment to the histone promoters almost completely depends on Asf1, HIR and acetylation of histone H3 at K56 (H3K56Ac), and is limited by Yta7, a bromodomain- and AAA-ATPase domain-containing boundary protein[31,32,35]. Although this role of Rtt106 at the histone promoters is relatively well-studied, whether Rtt106 plays a role at promoters other than those of histone genes remains unknown.

The yeast SWI/SNF complex is an ATP-dependent chromatin-remodelling complex, and it is composed of 12 subunits, 5 of which are fungal-specific[36]. SWI/SNF modulates nucleosome position to regulate the expression of a subset of genes including stress-responsive genes[20,37–40]. SWI/SNF localises to nucleosome-depleted regions (NDRs) in promoters and regulates the position of the transcription start site (TSS)-associated (or '+1') nucleosome critical for transcription initiation[20]. Recruitment of SWI/SNF to promoters can be mediated through sequence-specific transcription factors including Gcn4, Pho4, Swi5 and Hap4[41,42]. However, at the histone promoters, SWI/SNF is recruited by Rtt106, probably through direct physical interactions[33].

In this study we show that *S. cerevisiae* Rtt106 localises at subsets of specific promoters, binding particularly strongly to the promoters of the PDR network genes where it upregulates their expression exclusively through Pdr3. Consistently, loss of Rtt106 sensitised wild-type *S. cerevisiae* and a drug-resistant mutant (a GOF mutation in *PDR3*) to azole antifungal drugs. We further show that SWI/SNF acts as an essential activator of the PDR network genes for both basal and azole-induced expression, and therefore is essential for drug resistance of wild-type *S. cerevisiae*, and of a *pdr3*-GOF mutant. Lastly, we show that *C. glabrata* Rtt106 and SWI/SNF also localise at the promoter of the multidrug transporter gene *CgCDR1*, regulating its expression in response to an antifungal drug and conferring resistance to azole antifungals. Therefore, Rtt106 and SWI/SNF represent potential therapeutic targets for combating antifungal resistance of the pathogenic fungus *C. glabrata*.

## Results

**Rtt106 binds the PDR network gene promoters**. To examine which gene promoters are bound by Rtt106 genome-wide in *S. cerevisiae*, we performed ChIP-seq analysis. As expected[31,32], Rtt106 localised at the promoter of the histone gene pair *HTA1-HTB1* (Fig. 1a). Strikingly, we observed equally strong Rtt106 binding at the promoter of the multidrug transporter gene *PDR5* (Fig. 1b). The peak summit of Rtt106 binding to the *PDR5* promoter was observed at its four PDREs, which are recognised by transcription factors Pdr1 and Pdr3 (Fig. 1b). To identify further promoters that Rtt106 binds genome-wide, we calculated Rtt106 enrichment at all promoters, defined as the regions within 500 bp upstream of the TSS of each gene. Interestingly, 5 out of the top 10 Rtt106-binding promoters contain at least one PDRE (Fig. 1c). Furthermore, the Rtt106 peak summits coincided with the PDRE in such promoters e.g. *PDR5*, *PDR15*, *SNQ2* and *SPO24* (Fig. 2a, Rtt106 ChIP in WT). These results suggest that Rtt106 binds the regions containing PDRE in promoters of the PDR network genes.

Since PDRE is recognised by transcription factors Pdr1 and Pdr3[13], we tested if Pdr1 and Pdr3 are required for the recruitment of Rtt106 to the PDRE-containing promoters. Rtt106 binding to the PDRE-containing promoters was almost completely abolished in the absence of both Pdr1 and Pdr3 (Fig. 2a, Rtt106 ChIP in *pdr1Δ pdr3Δ*), indicating that recruitment of Rtt106 to the PDRE-containing promoters depends on Pdr1 or Pdr3 or both. Rtt106 binding to the

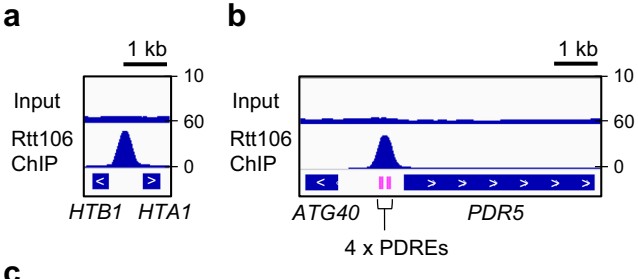

**Fig. 1 Rtt106 binds promoters containing PDRE features. a** Association of Rtt106 with the promoter of the histone gene pair *HTA1-HTB1*, validating ChIP-seq analysis of Rtt106-6HA performed in this study. *RTT106-6HA* cells in log phase were collected for ChIP-seq analysis. Input, sequence reads of the sonicated genomic DNA before ChIP. Rtt106 ChIP, sequence reads of ChIPed DNA with Rtt106-6HA. **b** ChIP-seq analysis of Rtt106 showing association of Rtt106 with the promoter of the *PDR5* gene. Positions of PDRE indicated by magenta lines. PDRE PDR responsive element. **c** The top 10 peaks of Rtt106 binding at promoter regions. Enrichment of Rtt106 at promoters was calculated as ChIP sequence reads normalised by Input sequence reads in a region spanning 500 bp upstream of the transcription start site of each gene.

**Top 10 peaks of Rtt106 binding at promoters**

| Gene promoters | Description | Feature in promoter | Rtt106 enrichment Log2 ratio (ChIP/Input) |
|---|---|---|---|
| *PDR5* | Multidrug transporter | PDRE | 4.8 |
| *HTA1* | Histone H2A | | 4.6 |
| *HTB1* | Histone H2B | | 4.4 |
| *SPO24* | Small protein involved in sporulation | PDRE | 3.2 |
| *SNQ2* | Multidrug transporter | PDRE | 2.9 |
| *PDR15* | Multidrug transporter | PDRE | 2.8 |
| *DFR1* | Dihydrofolate reductase | snoRNA | 2.4 |
| *PDR3* | Transcriptional activator of the PDR network | PDRE | 2.4 |
| *TFC3* | RNA pol III transcription initiation factor | Centromere | 2.3 |
| *YDR524C-B* | Putative protein | | 2.2 |

*HTA1-HTB1* promoter was not affected by the loss of Pdr1 and Pdr3, as expected since the *HTA1-HTB1* promoter does not contain a PDRE (Fig. 2b, Rtt106 ChIP in *pdr1Δ pdr3Δ*).

To further understand the mechanism of Rtt106 recruitment to PDRE-containing promoters, we tested for regulation by other factors involved in Rtt106 recruitment to histone gene promoters. We observed that the HIR histone chaperone, a crucial factor for recruitment of Rtt106 to the *HTA1-HTB1* promoter[31,32] (Fig. 2b, *hir1Δ*), contributes to Rtt106 recruitment to PDRE-containing promoters, since the loss of the HIR subunit Hir1 caused a reduction of Rtt106 binding (Fig. 2a, *hir1Δ*). We further observed that Yta7, a boundary protein that restricts Rtt106 binding at the histone promoters[31,32] (Fig. 2b, *yta7Δ*), limits Rtt106 binding also at the PDRE-containing promoters, as loss of Yta7 increased Rtt106 binding (Fig. 2a, *yta7Δ*). However, the *yta7Δ* mutation did not cause Rtt106 binding to spread from PDRE-containing promoters, unlike at the histone promoters where Rtt106 was observed to spread into regions normally occupied by Yta7 (Fig. 2a, b, compare our Rtt106 ChIP data with the published Yta7 ChIP data shown at bottom[29]). Yta7 may therefore limit Rtt106 binding through different mechanisms at the PDRE-containing and histone promoter sites. Taken together, our results show that HIR and Yta7 contribute to Rtt106 recruitment at the promoters of both the PDR network genes and the histone genes, but their specific effects differ somewhat in the two classes of promoters.

**Three distinct binding modes of Rtt106 at promoters**. To classify the types of promoters bound by Rtt106, we performed a clustering analysis of Rtt106-binding profiles obtained in a series of ChIP-seq analyses. Rtt106-binding promoters clustered into 9 groups (clusters 1–9) based on Rtt106 binding in WT and the effects of *pdr1Δ pdr3Δ*, *hir1Δ* and *yta7Δ* mutations (Fig. 2c, d and Supplementary Data 1). Interestingly, 19 out of 20 promoters in cluster 1 belong to PDR network genes (Fig. 2c–f), indicating that Rtt106 generally binds the promoters of the PDR network genes in a common manner that we designate 'Type A' i.e. fully dependent on Pdr1/3 and partially on HIR, and limited by Yta7. Note that not all PDRE-containing promoters were bound by Rtt106 (Supplementary Fig. 1a).

Cluster 2 contains promoters bound by Rtt106 dependent on HIR and limited by Yta7, but not dependent on Pdr1/3. This cluster included all histone gene pairs (except *HTA2-HTB2* where Rtt106 does not bind strongly), as well as *AGP1* and *ECL1* (Fig. 2b, e, f), indicating that Rtt106 may be recruited to the *AGP1* and *ECL1* promoter through the mechanism similar to that at histone promoters. We designated this binding type as 'Type B'.

Clusters 3–6 were very small groups, preventing any general statement about binding at these sites. In clusters 7 and 8, we found another type of Rtt106-binding promoters, where Rtt106 shows clear peaks (rather than broader signals) that do not depend on Pdr1, Pdr3 or HIR, and are not limited by the loss of Yta7 (Fig. 2e, 'peaks' in clusters 7 and 8, and Supplementary Fig. 1b and Supplementary Data 1). No common feature was found in these promoter regions. We designate such promoters with 'peaks' in cluster 7 and 8 as 'Type C' Rtt106-binding promoters. This third type contains some non-PDR transporter genes, e.g. *FUI1*, *CTR1*, *FTR1*, *PMA1*, *PHO87* and *PUT4*, but does not include any PDR genes. Gene ontology analysis of the 50 genes with these Type C promoters shows a significant enrichment of 'Cell Periphery' (*p*-value, 0.00015; 19 of 50 genes; Supplementary Fig. 1c). Rtt106 was also enriched at centromeres, *tRNA*, *snoRNA*, and gene bodies of highly expressed genes, mainly found in clusters 7–9 (Fig. 2e and Supplementary Fig. 1d and Supplementary Data 1). We did not consider promoters overlapping those Rtt106-binding features as 'Rtt106-bound promoters'.

Many Rtt106-bound promoters (61 out of 79 gene promoters) show regulated expression, either in response to environmental changes or during the cell cycle (Supplementary Data 2). We surmise that Rtt106 may generally bind a subset of gene promoters where expression is induced by specific cellular or environmental circumstances—such as PDR network genes, histone genes and other transporter genes.

We examined the relationship between Rtt106 binding and nucleosome organisation at promoters, by analysing nucleosome positioning and occupancy in MNase-seq datasets published by Rine and colleagues[29]. We observed that all three types of Rtt106-bound promoters show atypically wide NDRs (Fig. 2g), possibly reflecting the fact that inducible gene promoters are occupied by specific transcriptional regulators. At the histone promoters, *RTT106* deletion causes further reduction of nucleosome occupancy as reported previously[31] (Fig. 2g, Type B, compare green dashed line with solid line). Surprisingly, however, at the PDRE-containing (Type A) and the Type C promoters, Rtt106 removal did not alter the NDR profile, suggesting that at these promoters the histone chaperone activity of Rtt106 does not affect nucleosome positioning and occupancy at least under the tested conditions (Fig. 2g, compare *rtt106Δ* with WT).

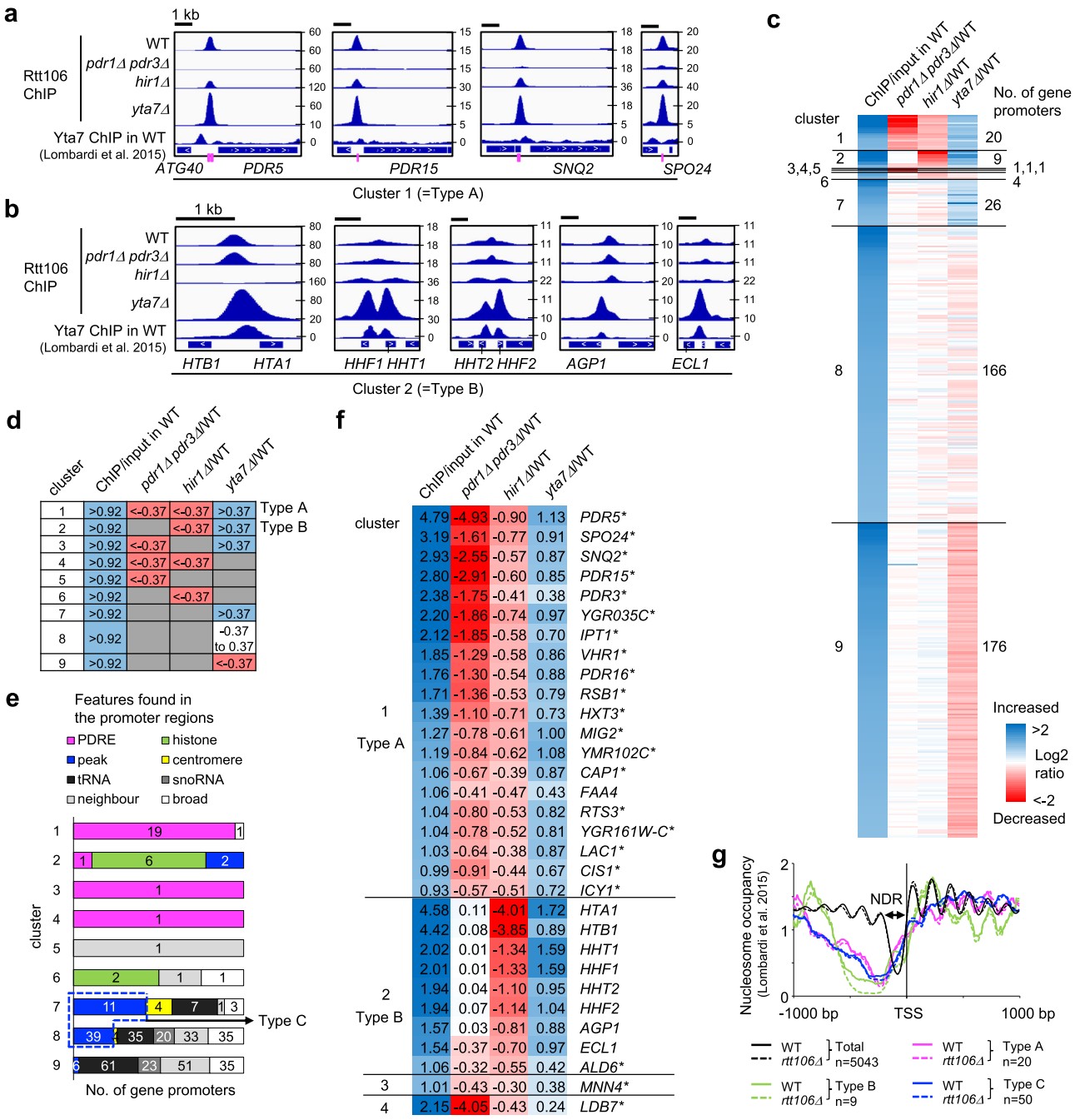

**Fig. 2 A series of ChIP-seq tests reveal three distinct modes of Rtt106 binding to promoters by clustering analysis. a** Specimen views of promoters where Rtt106 binding is dependent on Pdr1/3 and Hir1 and limited by Yta7 (falling into cluster 1, designated as Type A). Yta7 binding tested by ChIP-seq analysis[29] is shown alongside Rtt106 binding. Positions of PDRE indicated by magenta lines. **b** Specimen views of promoters where Rtt106 binding is independent of Pdr1/3, dependent on Hir1 and limited by Yta7 (falling into cluster 2, designated as Type B). ChIP-seq analyses of Rtt106 in WT, pdr1Δ pdr3Δ, hir1Δ and yta7Δ mutants are shown. **c** Clustering analysis of Rtt106 promoter association based on Rtt106 binding in WT and its changes in pdr1Δ pdr3Δ, hir1Δ and yta7Δ. Log2 fold enrichment of Rtt106 binding in WT at promoters (ChIP/input in WT) and log2 fold changes of Rtt106 binding in pdr1Δ pdr3Δ, hir1Δ and yta7Δ compared to that in WT (pdr1Δ pdr3Δ/WT, hir1Δ/WT and yta7Δ/WT, respectively) were calculated as described in Methods. **d** Criteria used for clustering in panel **c**. **e** Features found in promoters categorised into clusters 1–9. Numbers of gene promoters containing PDRE (PDR responsive element; magenta), centromere (yellow), tRNA (black) or snoRNA (dark grey), and those containing Rtt106 signals leaked from highly expressing neighbour genes ('neighbour'; light grey) are shown. Histone gene promoters ('histone'; green) were assessed separately. Other than the above features, numbers of promoters with Rtt106 peaks ('peak'; blue) or broad signals ('broad'; white) are shown. Promoters with 'peaks' in clusters 7 and 8 designated as Type C. **f** Gene promoters categorised into clusters 1–4. Log2 changes are shown as in **c**. Asterisks indicate gene promoters containing PDRE. **g** Nucleosome occupancy analysed by MNase-seq in WT and rtt106Δ (Lombardi et al. 2015[29]) aligned to transcription start site (TSS). Nucleosome-depleted region (NDR) in average of all gene promoters is indicated.

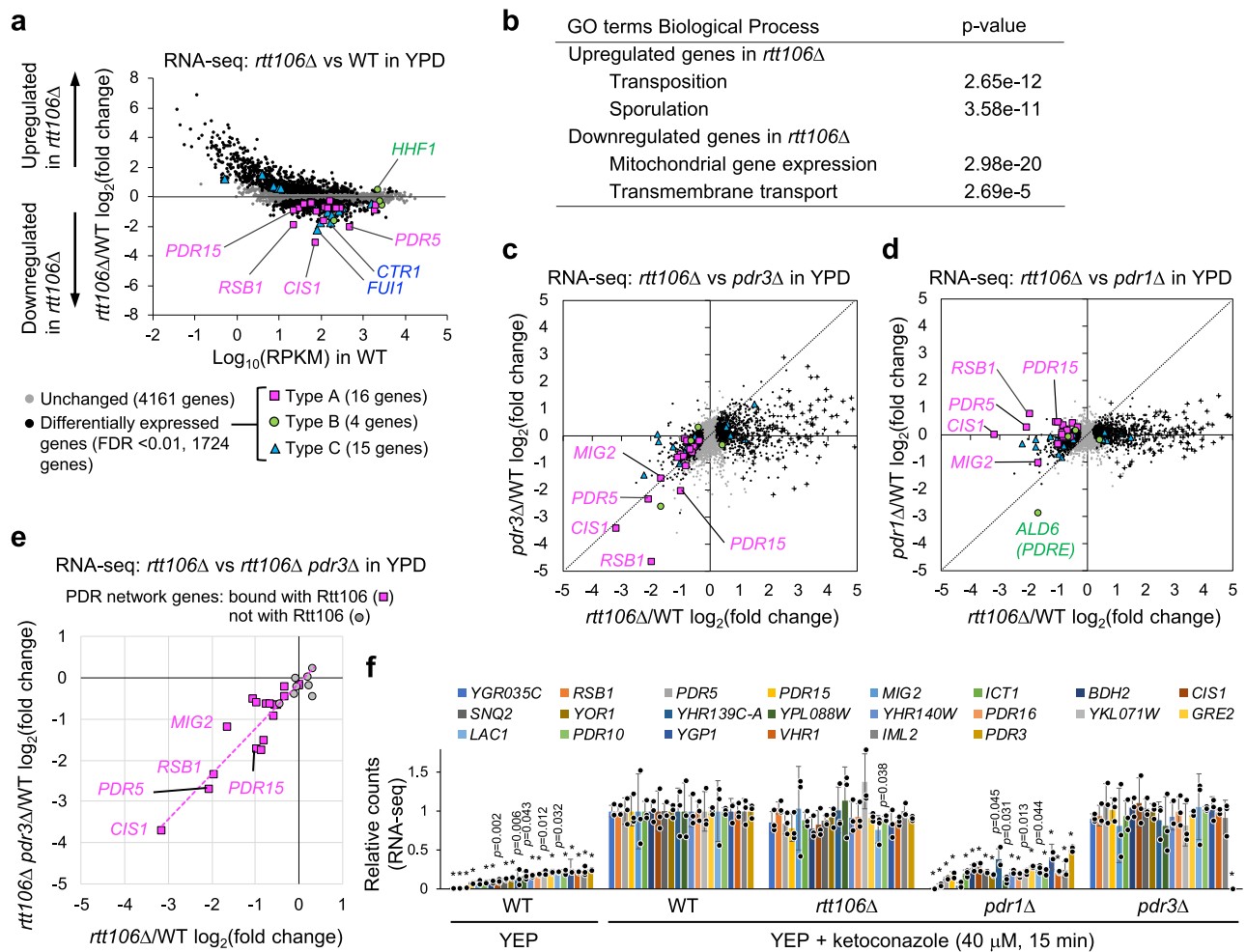

**Fig. 3 Rtt106 is essential for Pdr3-mediated basal expression of the PDR network genes. a** RNA-seq analysis of *rtt106Δ* versus WT grown in YPD. Read counts of transcripts in *rtt106Δ* divided by those in WT were plotted against RPKM (reads per kilobase of transcript per million mapped reads) in WT. Data shows mean of three biological replicates. Among 1724 differentially expressed genes (black circles, FDR < 0.01), genes in Type A (magenta squares), Type B (green circles) and Type C (blue triangles) as defined in Fig. 2 are highlighted. **b** Gene Ontology (biological process) analysis of upregulated genes (588 genes, FDR < 0.0001) and downregulated genes (266 genes, FDR < 0.0001) in *rtt106Δ*, compared to WT, in YPD. *P*-values were calculated by a hypergeometric distribution with Bonferroni Correction in GO Term Finder. **c** Correlation of changes in mRNA level of each gene in *pdr3Δ* with those in *rtt106Δ* grown in YPD. Colour coding as in panel **a**. Genes with low expression level in WT (RPKM < 1) are marked by +. **d** Comparison of changes in mRNA level of each gene in *pdr1Δ* with those in *rtt106Δ* grown in YPD. Colour cording used in panel **a** is used. Genes with low expression level in WT (RPKM < 1) are marked by +. **e** Correlation of changes in mRNA levels of the PDR network genes in *pdr3Δ rtt106Δ* with those in *rtt106Δ* grown in YPD. PDR network genes with promoters bound by Rtt106 are shown by magenta squares, and those not bound by Rtt106 by grey circles. **f** Normalised read counts of transcripts of the PDR network genes relative to those in WT treated transiently by ketoconazole in YEP. The 22 PDR network genes whose transcripts increased more than 4-fold by ketoconazole treatment in WT are shown. Data are presented as mean values ± SD. Statistical significance determined by one-way ANOVA with post-hoc Tukey HSD tests, compared to WT treated with ketoconazole. *$p = 0.001$. **c–f** Data shows mean of three biological replicates.

**Rtt106 upregulates Pdr3-mediated expression of the PDR network genes.** Rtt106 regulates the expression of the histone genes[31,32], so we tested if it also regulates expression at these newly identified binding sites. We performed RNA-seq analysis of total RNA prepared from WT and *rtt106Δ* cells in log phase growth in rich medium (YPD). Loss of Rtt106 caused a significant reduction of mRNA levels of the majority of the PDR network genes whose promoters it binds (16 out of 20 genes in Type A), with a particular reduction in *PDR5*, *CIS1* and *RSB1* expression (Fig. 3a and Supplementary Fig. 2a and Supplementary Data 3). These results suggest that Rtt106 generally upregulates the expression of the PDR network genes, in contrast to its repressive role in histone gene expression. The effect of Rtt106 loss on mRNA levels of the genes classified in Type C was not uniform.

mRNA levels of some of those genes (10 out of 50 genes) were significantly reduced, including the uridine permease *FUI1* and the copper transporter *CTR1* (Fig. 3a and Supplementary Data 3). Gene ontology analysis of the downregulated genes in *rtt106Δ* show significant enrichment of 'Transmembrane transport' genes (Fig. 3b), and 8 out of the 20 most reduced mRNA in *rtt106Δ* correspond to transporter genes (Supplementary Fig. 2a). Rtt106 may therefore be generally important for the expression of transporter genes—even though we did not identify Rtt106 bound at all the gene promoters affected, suggesting some effects could be indirect.

Regarding histone gene expression, in our experiments only *HHF1* mRNA was significantly increased in *rtt106Δ* (Fig. 3a and Supplementary Data 3), unlike previous reports which found

additional histone genes affected[31,32]. This difference may reflect the fact that we prepared RNA from asynchronous cells rather than cells at specific cell-cycle stages[31,32].

Remarkably, loss of Rtt106 caused an increase in mRNA levels for most genes that normally show low expression (Fig. 3a) including meiosis genes, for example, genes related to sporulation (Fig. 3b) that are suppressed in vegetative growth. This may reflect increased transcription initiation that could be caused by defective histone deposition or increased nucleosome spacing in rtt106Δ[26,27,29]. No obvious correlation of genes whose expression increased in rtt106Δ and Rtt106 binding at those genes was observed. We suspect that mildly deregulated transcription initiation occurs fairly ubiquitously in rtt106Δ, resulting in a readily apparent fold increase of mRNA with low abundance, but a less obvious effect on mRNA with a high level of expression. An increase in mRNA levels of 'Transposition' genes such as retrotransposons (Fig. 3b) is probably linked to the effect of Rtt106 as a Regulator of Ty1 Transposition (RTT)[43], the phenotype for which Rtt106 was originally identified.

To test if Rtt106 is involved in transcription directed by either Pdr1 or Pdr3 or both, we compared effects on mRNA levels in rtt106Δ with those in pdr1Δ or pdr3Δ (grown in YPD without the addition of any drug). Almost all PDR network genes whose mRNA levels were reduced in rtt106Δ showed a similar reduction in pdr3Δ; however, no reduction was observed in the mRNA for most such genes in pdr1Δ (Fig. 3c, d and Supplementary Data 3). These results suggest that Pdr3, not Pdr1, is the main transcription factor responsible for the expression of the PDR network genes at basal levels and that Rtt106 acts with Pdr3 in these conditions. Consistent with that notion, we further observed that deletion of PDR3 in the rtt106Δ background had no additive effect on mRNA changes when compared to the single rtt106Δ mutant (Fig. 3e and Supplementary Table 1). These results suggest that Pdr3-mediated transcription of the PDR network genes is almost completely dependent on Rtt106. Our finding that Pdr3, and not Pdr1, is the main transcription factor responsible for basal expression of the PDR network genes is consistent with previous genome-wide transcriptome analysis[44] and other published papers (listed in Supplementary Data 4). A number of papers, however, have arrived at different conclusions (also listed in Supplementary Data 4), in finding Pdr1 as the main contributor[12] or suggesting an equal contribution of Pdr1 and Pdr3[11]. The differences in behaviour between Pdr1 and Pdr3 could be due to differences in genetic background, a technique used and/or their combination.

As expected, neither pdr1Δ nor pdr3Δ mutations caused increased mRNA from genes with low expression levels ('+' symbols in Fig. 3c, d), consistent with the effect on these genes in rtt106Δ being due to a general histone occupancy defect, occurring independently of Pdr3. mRNA levels of non-PDR genes that are decreased in rtt106Δ also tended to decrease in pdr3Δ (Fig. 3c and Supplementary Data 3). We suspect that this may represent an indirect consequence of PDR network gene dysregulation, since it is unlikely that such genes are directly controlled by Pdr3.

**Pdr1, but neither Rtt106 nor Pdr3, is essential for drug-induced PDR network gene expression.** Expression of the PDR network genes can be induced by many substances, including the antifungal azole drug ketoconazole[7]. We tested if Rtt106, Pdr1 and Pdr3 are required for ketoconazole-induced expression of the PDR network genes. We used an established protocol[7] in which glucose-starved cells (which have low PDR5 mRNA levels) were transiently treated with ketoconazole, causing rapid induction of PDR5 expression (Supplementary Fig. 2b). YPD was not used for these induction experiments since the azole antifungals tested did not induce PDR5 expression in YPD (Supplementary Fig. 2c). We used ketoconazole to test drug-induced expression since fluconazole did not induce PDR5 expression (Supplementary Fig. 2b). The mRNA levels of 22 genes, all with PDRE in their promoters, were increased more than four-fold in response to ketoconazole in WT (Fig. 3f). We found that PDR1 is essential for the ketoconazole-induced, transient expression of these 22 genes, but neither PDR3 nor RTT106 is required (Fig. 3f).

Overall, the results in Fig. 3 suggest that Rtt106 and Pdr3 are essential for basal expression of the PDR network genes during growth without antifungal agents, while Pdr1 mediates their azole-induced, transient expression.

**Rtt106 confers antifungal resistance via Pdr3-mediated expression of PDR5.** We next focused on the role of Rtt106 in regulating PDR network gene PDR5, since Pdr5 is the main multidrug transporter critical for resistance to azole antifungal drugs. In addition to RNA-seq (Fig. 3), Northern blot analyses confirmed that Rtt106 acts with Pdr3 to mediate basal expression of PDR5, but Rtt106 is not essential for Pdr1-mediated transient expression of PDR5 in response to ketoconazole (Supplementary Fig. 3a–d). Consistently, ketoconazole treatment did not cause a significant increase in the recruitment of Rtt106 to the PDR5 promoter (Supplementary Fig. 3e).

We further found using a luciferase expression assay that Rtt106 is required for activating the PDR5 promoter (Fig. 4a), and therefore for maintaining Pdr5 protein levels under non-drug-treated conditions (western blot analysis; Fig. 4b). Consistently, loss of RTT106 sensitised S. cerevisiae to azole antifungal drugs, to an extent similar to pdr3Δ (Fig. 4c and Supplementary Fig. 3f). We further observed that pdr3Δ and rtt106Δ are sensitive to high doses of ketoconazole, while pdr1Δ is sensitive only to low doses but not to high doses (Supplementary Fig. 3f). This finding suggests that Pdr3 and Rtt106 are more critical for resistance to continuous high doses of azole antifungals than Pdr1; and that Pdr1 function alone cannot mediate resistance to high doses at the condition tested. Although these are unexpected results since Pdr1 is critical for drug-induced expression of PDR5, this observation might reflect the fact that PDR3 undergoes auto-activation of its own transcription[45], while PDR1 does not. Overall, these results suggest that Rtt106 regulates basal expression of PDR5 via Pdr3 and also mediates drug resistance. Note that it is still unknown whether basal expression of PDR5 regulated by Rtt106 directly leads to drug resistance, and whether Rtt106 also affects drug-induced PDR5 expression in cells grown long-term with the drug.

To understand why Rtt106 regulates Pdr3-mediated, but not Pdr1-mediated, transcription of PDR5, we tested if Rtt106 recruitment to the PDR5 promoter depends solely on Pdr3. Strikingly, deletion of PDR3 caused almost complete loss of Rtt106 recruitment to the PDR5 promoter, while deletion of PDR1 actually increased recruitment of Rtt106 to the PDR5 promoter (Fig. 4d). Rtt106 recruitment to the PDR5 promoter, therefore, depends on Pdr3, and not Pdr1. Increased recruitment of Rtt106 to the PDR5 promoter in pdr1Δ may reflect an increase of Pdr3 at the PDR5 promoter in the absence of Pdr1. These results suggest that Pdr3 recruits Rtt106 to the PDR5 promoter to upregulate its transcription.

Conversely, Rtt106 is not essential for Pdr3 recruitment to the PDR5 promoter, as Pdr3 is still recruited there in the absence of Rtt106, as shown by ChIP-qPCR using a tagged PDR3-13Myc strain confirmed as retaining protein function (Supplementary Fig. 4a–d). However, we observed ~50% reduction of Pdr3 binding to the PDR5 promoter in rtt106Δ, compared to WT,

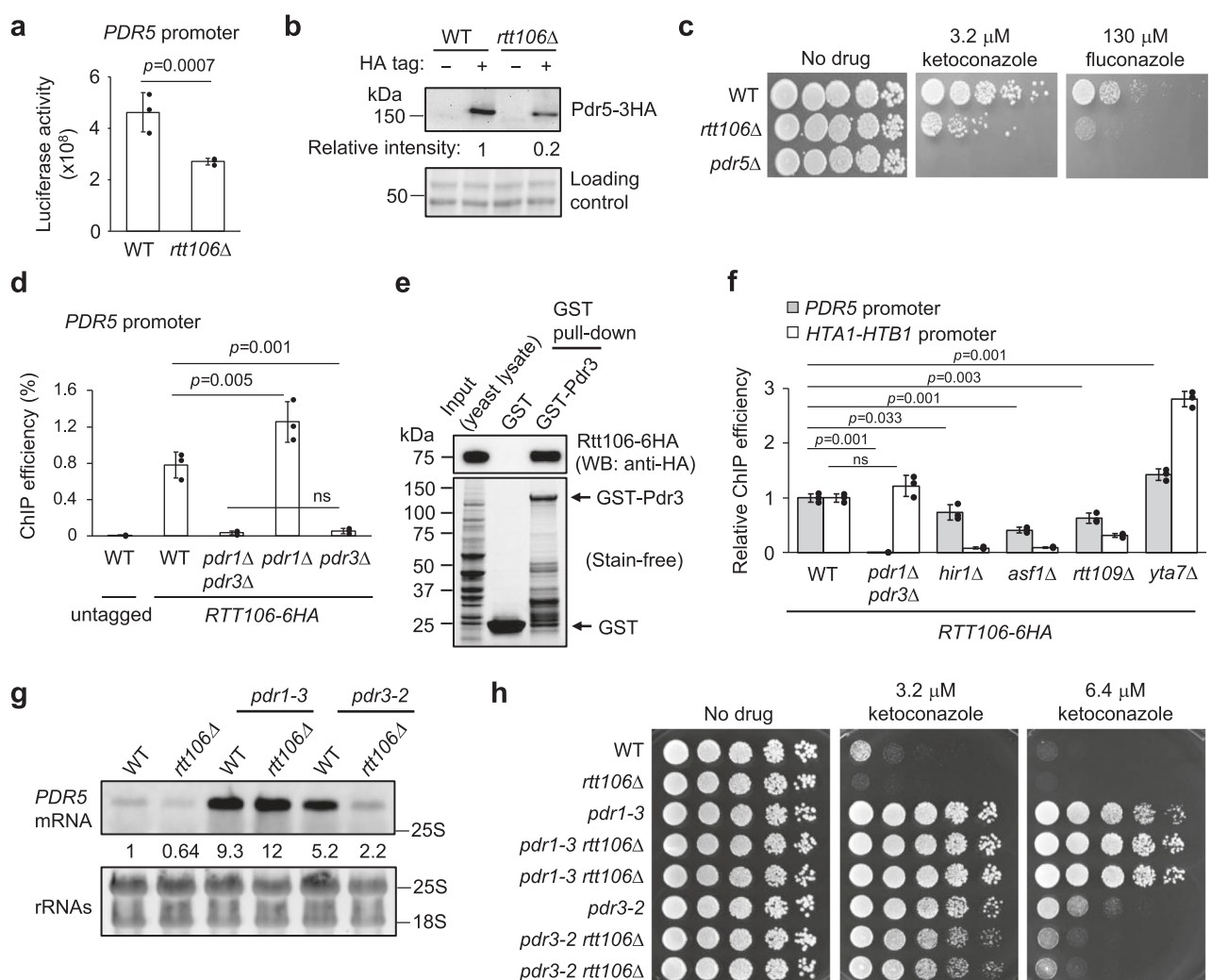

**Fig. 4 Rtt106 mediates azole antifungal resistance through Pdr3-dependent *PDR5* transcription. a** Promoter activity of *PDR5* in WT and *rtt106Δ* assessed by luciferase assay. Data are presented as mean values ± SD ($n = 3$ biological replicates). Statistical analysis performed by an unpaired *t*-test (two-tailed). **b** Western blot analysis of Pdr5 in WT and *rtt106Δ*. HA-tagged Pdr5 expressed from its endogenous locus was detected by anti-HA antibody. For loading control, whole proteins were detected by the Stain-free technology. **c** Sensitivity of WT, *rtt106Δ* and *pdr5Δ* to azole antifungal drugs. Five-fold dilutions of WT, *rtt106Δ* and *pdr5Δ* cells were spotted on YPD and YPD containing ketoconazole or fluconazole and incubated at 30 °C for 3 days. **d** ChIP-qPCR analysis testing if Rtt106 binding at the *PDR5* promoter is dependent on Pdr1 and/or Pdr3. ChIP efficiency, the recovery of ChIPed DNA relative to the amount of input. Data are presented as mean values ± SD ($n = 3$ biological replicates). Statistical significance determined by one-way ANOVA with post-hoc Tukey HSD test. ns, $P > 0.05$. **e** GST pull-down analysis shows GST-Pdr3 interacts with Rtt106-6HA. Yeast cell lysate containing Rtt106-6HA was incubated with GST-Pdr3 bound to glutathione-Sepharose beads. WB western blot. **f** ChIP-qPCR analyses assessing dependence of Rtt106 binding to *PDR5* (grey) and *HTA1-HTB1* (white) promoters on Pdr1, Pdr3, Hir1, Asf1, Rtt109 and Yta7. ChIP efficiencies relative to WT are shown. Data are presented as mean values ± SD ($n = 3$ biological replicates). ns, $P > 0.05$, one-way ANOVA with post-hoc Tukey HSD tests. **g** Northern blot analysis of *PDR5* mRNA in indicated strain backgrounds in YPD. Intensities relative to *PDR5* mRNA in WT are shown below the *PDR5* blot. rRNAs, loading control, detected by SYBR Gold. The positions of 25 S and 18 S rRNAs are shown. **h** Sensitivity to ketoconazole of indicated mutant combinations. Five-fold dilutions of indicated strains were spotted on YPD and YPD containing ketoconazole and incubated at 30 °C for 3 days. Two independent isolates of *pdr1-3 rtt106Δ* and *pdr3-2 rtt106Δ* strains shown.

possibly caused by the reduction of protein and mRNA levels of *PDR3* in *rtt106Δ* (Supplementary Fig. 4d, e and Supplementary Data 3). While not ruling out some direct contribution of Rtt106 to Pdr3 recruitment, these results suggest clearly that Rtt106 regulates the expression of *PDR3*, one of the PDRE-containing genes. The reduction of Pdr3 protein level and its binding to the *PDR5* promoter in *rtt106Δ* may contribute to the reduction of *PDR5* expression in *rtt106Δ*. Our results nonetheless indicate that Rtt106 regulates *PDR5* expression by acting directly at the *PDR5* promoter (and not only indirectly through regulating *PDR3* expression) as a ~50% reduction of Pdr3 binding at the *PDR5*

promoter in *rtt106Δ* cannot alone explain why *rtt106Δ* and *pdr3Δ* mutants have similar effects on *PDR5* expression.

We next tested the physical interaction of Rtt106 with Pdr3 by an in vitro GST pull-down analysis and observed that Rtt106 shows a higher affinity for GST-Pdr3 than GST alone, when testing cell lysate from the yeast with HA-tagged Rtt106 (Fig. 4e), suggesting that Rtt106 binds Pdr3 in this condition. However, His-Rtt106 purified from *E. coli* expression system was not co-precipitated with GST-Pdr3 (Supplementary Fig. 4f), implying there may be another molecular requirement for binding, or possibly that recombinant Rtt106 is not properly

folded to bind Pdr3 in the condition tested. How Rtt106 binds Pdr3 remains to be elucidated.

We tested if factors important for recruiting Rtt106 to the histone promoters also contribute to its recruitment to the *PDR5* promoter. We found that histone chaperone Asf1 and histone acetyltransferase Rtt109 (in addition to HIR and Yta7) also contribute to Rtt106 recruitment to the *PDR5* promoter, although their contributions at *PDR5* are less than at the *HTA1-HTB1* histone promoter (Fig. 4f). Asf1 might affect Rtt106 recruitment to the *PDR5* promoter indirectly through Pdr3 since Pdr3 binding to the *PDR5* promoter is reduced in *asf1Δ* (Supplementary Fig. 4d).

We then tested if Rtt106 mediates *PDR5* overexpression caused by GOF mutations in *PDR1* and *PDR3*. As expected, loss of Rtt106 caused the reduction of *PDR5* mRNA in a *pdr3*-GOF (*pdr3-2*) mutant, but not in a *pdr1*-GOF (*pdr1-3*) mutant (Fig. 4g). Consistently, loss of Rtt106 sensitised *pdr3*-GOF, but not *pdr1*-GOF, to ketoconazole (Fig. 4h). Rtt106 is therefore critical to mediate hyperactivation of Pdr3. These selective effects on *pdr3*-GOF also suggest that the effect of Rtt106 loss on *PDR5* expression is not simply caused indirectly by global change in chromatin structure due to loss of Rtt106. We were unable to test the contribution of Rtt106 to mitochondrial dysfunction-mediated drug resistance[16], since we have not observed such phenotype reproducibly in the strain background used (Supplementary Fig. 5a, b).

**SWI/SNF is recruited to the *PDR5* promoter dependent on Pdr1/3.** To further investigate *PDR5* gene regulation, we sought to identify proteins assembled on the *PDR5* promoter. To capture proteins assembled at the *PDR5* promoter in a near-native in vivo context, we utilised a minichromosome isolation technique[46] followed by SILAC-based quantitative proteomics (Fig. 5a, b and Supplementary Fig. 6a). In this method, we purified from *S. cerevisiae* short circular chromosomes containing the *PDR5* promoter and eight lactose operators (*lacO*) by immunoprecipitating FLAG-tagged Lac Repressor (LacI) (Fig. 5a). Proteins bound to the *PDR5* promoter will be enriched on the *PDR5* promoter-containing minichromosome (PDR5pro), compared to a control minichromosome (Empty). We successfully identified *PDR5* promoter-binding proteins including transcription factors Pdr1 and Pdr3, and the RNA polymerase II Mediator complex that binds Pdr1 and Pdr3[7,47], validating this approach for capturing promoter-binding proteins in a near-native context (Fig. 5b and Supplementary Data 5). Rtt106 enrichment on the *PDR5* promoter was confirmed by western blot analysis, although it was not identified by mass spectrometry (Supplementary Fig. 6a). Interestingly, we identified the SWI/SNF ATP-dependent chromatin-remodelling complex and the transcriptional regulatory protein Ume6 (Fig. 5b and Supplementary Data 5) as enriched on the *PDR5* promoter. Ume6 likely binds to the DNA sequence 5′-TTGCCGCCGA-3′ located 580 bp upstream of the *PDR5* ATG, which is similar to the Ume6 recognition sequence 5′-TAGCCGCCGA-3′.

We further found that SWI/SNF recruitment to the *PDR5* promoter is dependent mainly on Pdr3 (Fig. 5c, d). Pdr1 might contribute to the recruitment of SWI/SNF to the *PDR5* promoter since residual binding of SWI/SNF was observed in the single *pdr3Δ* mutant, compared to the double *pdr1Δ pdr3Δ* mutant (Fig. 5c, d). Consistent with this idea, SWI/SNF recruitment to the *PDR5* promoter increased in response to ketoconazole, which stimulates only Pdr1 in the condition tested (Fig. 5e). Furthermore, GST pull-down analyses showed that both Pdr1 and Pdr3 bind purified SWI/SNF (Fig. 5f, g and Supplementary Fig. 6b). Interestingly, a few peptides from Pdr1 were identified in the

immunopurified fraction of SWI/SNF, by mass spectrometry (Supplementary Fig. 6c, highlighted in red), providing further supporting evidence for the interaction of SWI/SNF with Pdr1. Note that the checkpoint kinase Mec1 (that binds SWI/SNF[48]) and its binding partners were also identified in this fraction (Supplementary Fig. 6c, highlighted in yellow), and therefore the involvement of Mec1 in the SWI/SNF-Pdr1/3 interaction is not ruled out.

**SWI/SNF upregulates the expression of *PDR5* to mediate antifungal resistance.** We next explored if SWI/SNF is required for the expression of *PDR5*. Our RNA-seq analysis showed that *PDR5* mRNA was greatly decreased in the absence of Snf2, a catalytic subunit of SWI/SNF, when grown in YPD (Fig. 5h). Interestingly, Snf2 also contributes to ketoconazole-induced, transient expression of PDR network genes including *PDR5* (Fig. 5i, j). These results suggest that SWI/SNF is an essential upregulator of the PDR network genes both for basal and drug-induced expression.

As expected, the protein level of Pdr3 in *snf2Δ* is reduced, compared to WT (Supplementary Fig. 4e), consistent with the reduction of mRNA level of *PDR3* in *snf2Δ* (Supplementary Table 2). Concomitantly, Pdr3 binding to the *PDR5* promoter is reduced in *snf2Δ*, compared to WT (Supplementary Fig. 4d). Taken together, these results and the evidence of SWI/SNF binding to the *PDR5* promoter suggest that SWI/SNF regulates *PDR5* expression by acting directly at the *PDR5* promoter as well as through regulating gene expression of the transcription factor *PDR3*.

Consistent with an essential role of Snf2 in upregulating *PDR5* expression, *snf2Δ* is highly sensitive to azole antifungals (Fig. 5k and Supplementary Figs. 3f and 6d). Loss of all other SWI/SNF subunits tested also sensitised *S. cerevisiae* to azole antifungals, except for *snf11Δ* and *swp82Δ* in which *PDR5* mRNA levels were not reduced[44] (Fig. 5k and Supplementary Fig. 6d, e). We further observed that loss of Snf6 (a fungal-specific subunit of SWI/SNF) reduced *PDR5* mRNA in the hyperactive *pdr3*-GOF mutant (*pdr3-2*) and sensitised it to ketoconazole (Fig. 5l, m). Loss of Snf6 did not reduce *PDR5* mRNA in the hyperactive *pdr1*-GOF (*pdr1-3*) grown in YPD, but combining *pdr1*-GOF and *snf6Δ* caused a major growth defect even in the absence of antifungal drugs (Fig. 5l, n) through an unknown mechanism. Taken together, these results suggest that SWI/SNF is an essential factor mediating resistance to azole antifungals through upregulating *PDR5* expression, and therefore represents a potential therapeutic target to combat antifungal resistance.

**Cooperation of SWI/SNF and Rtt106 at the PDR network gene promoters.** Since both Rtt106 and SWI/SNF bind the *PDR5* promoter, we next examined if these factors colocalise at other promoters, by combining the published promoter-binding data for SWI/SNF[20] with our Rtt106 ChIP-seq data (Fig. 6a, b). SWI/SNF binds strongly to 16 of the 20 Rtt106-bound promoters classified as Type A (Fig. 6b and Supplementary Data 6), indicating that SWI/SNF colocalises with Rtt106 at the majority of the PDR network gene promoters, and suggesting that these two factors are likely to cooperatively regulate expression of the PDR network genes. Consistently, changes in mRNA levels of genes in Type A in *snf2Δ* are similar to those in *rtt106Δ* (Fig. 6c and Supplementary Table 2). Rtt106 and SWI/SNF factors also tended to colocalise at Type B promoters. In contrast, no clear correlation between SWI/SNF and Rtt106 binding was observed for Type C genes (Fig. 6b, c). The possibility that Rtt106 and SWI/SNF cooperate is further supported by their physical interaction (Fig. 6d)[33]. Note though that Rtt106 and SWI/SNF do not always

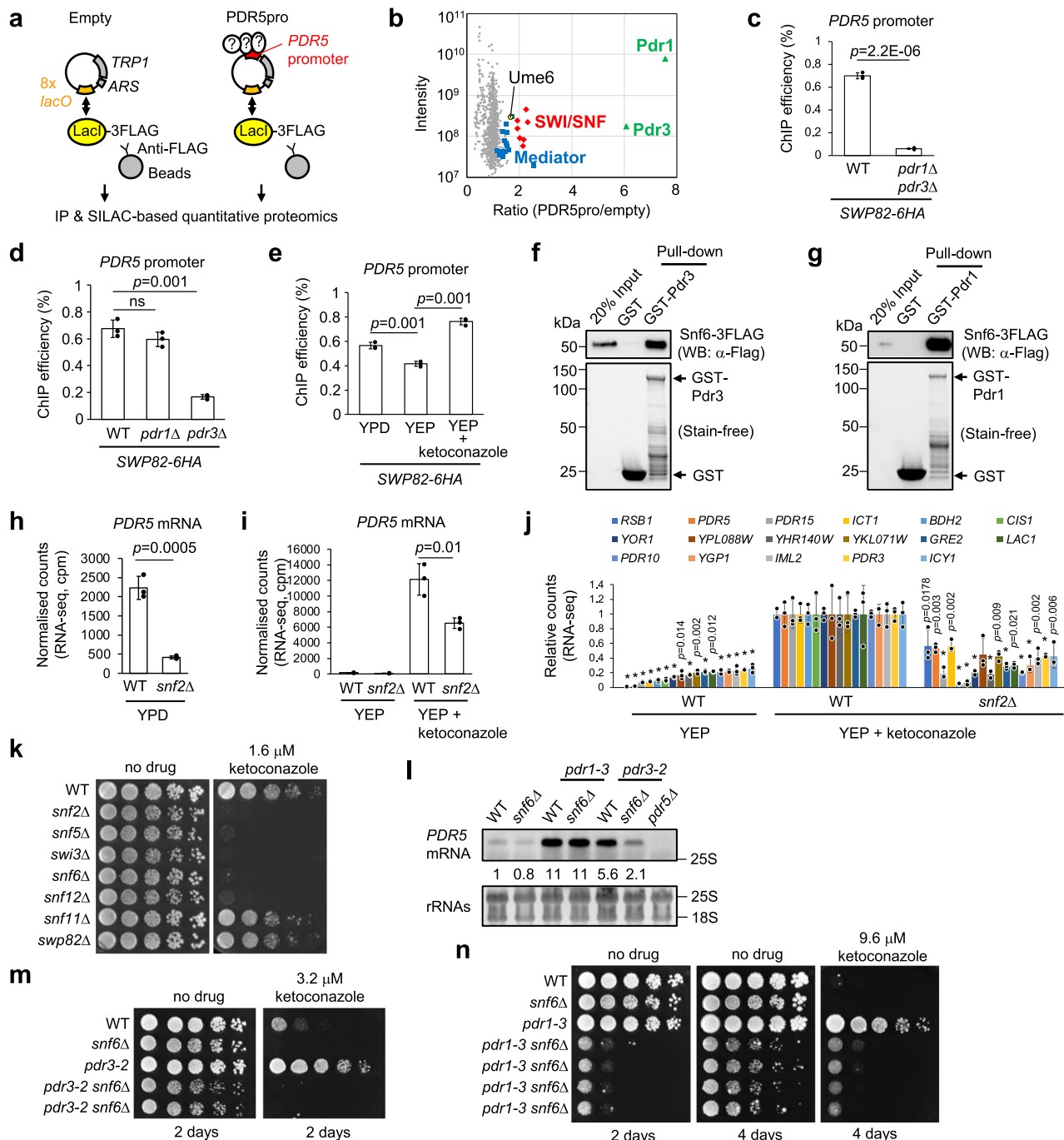

**Fig. 5 SWI/SNF is recruited to the *PDR5* promoter dependent on Pdr1/3 to upregulate its transcription and confer azole antifungal resistance.**
**a** Minichromosome purification scheme followed by SILAC-based quantitative proteomics. ARS replication origin. *TRP1* an auxotrophic selection marker. IP immunoprecipitation. **b** Ratios of proteins associated with *PDR5* promoter minichromosome compared to control minichromosome. Red diamonds, subunits of SWI/SNF; blue squares, subunits of Mediator; green rectangle, Pdr1 or Pdr3; yellow circle, Ume6. Proteins quantified by at least 5 ratio counts were shown (*n* = 1 experiment). **c**–**e** ChIP-qPCR analyses of the Swp82 subunit of SWI/SNF at *PDR5* promoter, in WT and *pdr1Δ pdr3Δ* (**c**), *pdr1Δ* and *pdr3Δ* (**d**) in YPD and in response to ketoconazole (**e**). Data are presented as mean values ± SD (*n* = 3 biological replicates). Unpaired *t*-test, two-tailed (**c**); one-way ANOVA with post-hoc Tukey HSD tests (**d**, **e**). ns, *P* > 0.05. **f**, **g** GST pull-down analyses show that the SWI/SNF complex purified from yeast cells interacts with GST-Pdr3 (**f**) and GST-Pdr1 (**g**). **h**, **i** Normalised counts of *PDR5* mRNA extracted from RNA-seq data of WT and *snf2Δ* grown in YPD (**h**) and treated with ketoconazole (**i**). Data are presented as mean values ± SD (*n* = 3 biological replicates). Unpaired *t*-tests (two-tailed). **j** Normalised read counts of transcripts of the PDR network genes relative to those in WT treated transiently by ketoconazole in YEP. PDR network genes whose transcripts increased more than four-fold on ketoconazole treatment in WT are shown. Genes whose expression is already higher in *snf2Δ* than WT in YEP without ketoconazole are excluded. Data are presented as mean values ± SD (*n* = 3 biological replicates). One-way ANOVA with post-hoc Tukey HSD tests, compared to WT treated with ketoconazole. \**p* = 0.001. **k** Sensitivity to ketoconazole of deletion mutants of the SWI/SNF subunits. **l** Northern blot analysis of *PDR5* mRNA in indicated mutants in YPD. The intensities relative to *PDR5* mRNA prepared from WT are shown below the *PDR5* blot. rRNAs, loading control. The positions of 25 S and 18 S rRNAs are shown. **m**, **n** Sensitivity to ketoconazole of WT and *snf6Δ* in hyperactive *pdr3-2* (**m**) and *pdr1-3* (**n**) mutant backgrounds. Two and four independent isolates of *pdr3-2 snf6Δ* and *pdr1-3 snf6Δ* strains used, respectively.

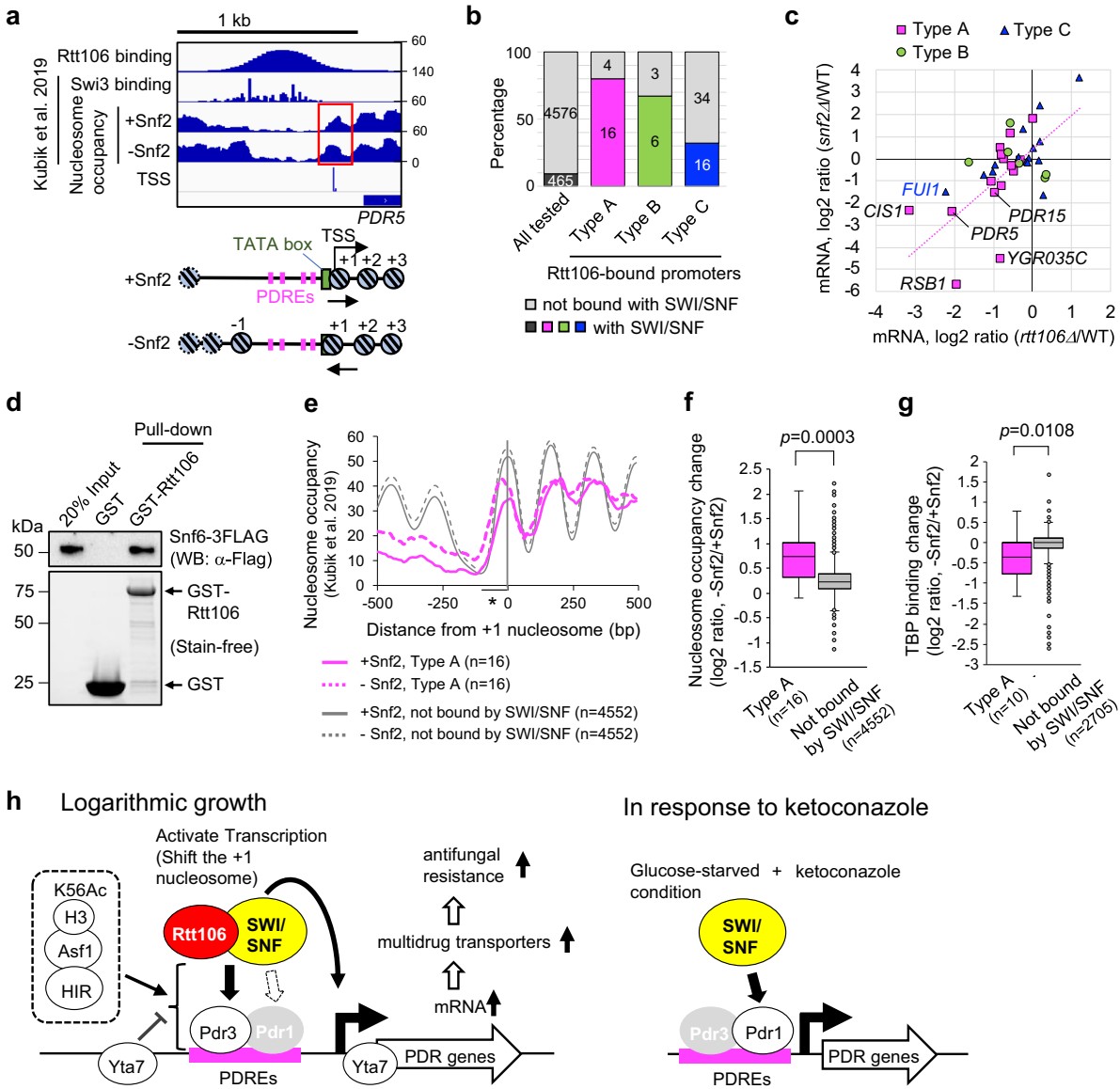

**Fig. 6 SWI/SNF colocalises with Rtt106 and regulates TSS-associated nucleosome position at PDR network gene promoters. a** Nucleosome positioning at the *PDR5* promoter in the presence and absence of Snf2. Binding of the Swi3 SWI/SNF subunit (ChEC-seq) and nucleosome occupancy (MNase-seq) at the *PDR5* promoter were extracted from published datasets[20]. TSS TSS-seq extracted from Malabat et al. 2015[49]. Snf2 depletion resulted in shift of the +1 nucleosome at the *PDR5* promoter (red rectangle). **b** Percentages of gene promoters strongly bound by SWI/SNF and those by Rtt106. The list of gene promoters strongly bound with SWI/SNF was obtained from Kubik et al. 2019[20]. Promoters bound by Rtt106 were defined as in Fig. 2. **c** Changes in mRNA levels of genes with promoters bound by both SWI/SNF and Rtt106 (as in **b**), and correlation of *snf2Δ* and *rtt106Δ* effects in YPD. Magenta dashed line shows trend for Type A. Data represents means of three biological replicates. **d** GST pull-down analysis showing GST-Rtt106 interacts with SWI/SNF purified from yeast cells. **e** Nucleosome occupancies at Type A promoters bound by SWI/SNF (magenta lines) and promoters not bound by SWI/SNF (grey lines), and those in the absence of Snf2 (dashed lines). Datasets in Kubik et al. 2019[20] were utilised. Asterisk, region analysed in **f** and **g**.
**f**, **g** Nucleosome occupancy change (**f**) and TBP binding change (**g**) caused by loss of Snf2 at Type A promoters bound by SWI/SNF. Normalised reads in regions spanning −100 to 0 bp from the dyad of the +1 nucleosome (indicated by asterisk in **e**) were extracted from published datasets[20]. For analysis of TBP binding change, promoters with low TBP binding (<50 normalised reads) were eliminated. Boxplots indicate median (middle line), interquartile range (box), outliers denoted by points greater than ±1.5 × interquartile range (single points) and the rest of the data distribution (whiskers). Mann–Whitney tests (two-sided). **h** Proposed roles of Rtt106 and SWI/SNF in transcription of the PDR network genes in *S. cerevisiae* (S288C). Rtt106 and SWI/SNF are recruited to the PDR network gene promoters dependent on Pdr3, to mediate Pdr3-dependent transcription during logarithmic growth phase, while SWI/SNF upregulates their Pdr1-dependent expression in response to ketoconazole.

colocalise at promoters (Supplementary Fig. 7a), suggesting a specific requirement for their colocalisation at a subset of promoters.

To understand how SWI/SNF regulates the expression of the PDR network genes, we explored if SWI/SNF regulates the position of the TSS-associated +1 nucleosome, which is critical

for transcription initiation. We utilised the published datasets of nucleosome occupancy[20] and TSS[49] and observed that in the absence of Snf2 the +1 nucleosome is shifted towards the TATA-box at the *PDR5* promoter (Fig. 6a, red rectangle). Snf2 loss had a similar effect at other PDR network gene promoters bound by both SWI/SNF and Rtt106 (Fig. 6e, compare dashed and solid

magenta lines), while no obvious change was observed at promoters not bound by SWI/SNF (Fig. 6e, grey lines). To test for statistical significance, we analysed nucleosome occupancy change caused by loss of Snf2 in regions spanning −100 to 0 bp from the dyad of the +1 nucleosome (regions indicated by an asterisk in Fig. 6e). We observed that increase of nucleosome occupancy caused by loss of Snf2 in those regions is significantly higher at the PDR network gene promoters bound by SWI/SNF and Rtt106 than at promoters not bound by SWI/SNF (Fig. 6f). These results suggest that SWI/SNF regulates the position of the TSS-associated +1 nucleosome at promoters of the PDR network genes.

The changes in nucleosome occupancy and position caused by the loss of Snf2 likely reduce the accessibility of the TATA-box and therefore reduce its transcription (Fig. 6a lower cartoons). Consistent with this idea, the binding of the TATA-box-binding protein TBP to the PDR network gene promoters is significantly reduced in the absence of Snf2, compared to that of promoters not bound by SWI/SNF (Fig. 6g and Supplementary Fig. 7b). These results are consistent with a role for SWI/SNF in directing the position of the +1 nucleosome at the PDR5 promoter to activate its transcription. In contrast, loss of Rtt106 does not change the steady-state level of nucleosomes at the PDR5 promoter (Supplementary Fig. 7c)[29], leaving the possibility that at the PDR5 promoter Rtt106 may regulate nucleosome turnover, an effect not observable by MNase-seq nucleosome mapping.

Taken together, we propose that during logarithmic growth Rtt106 and SWI/SNF bind to promoters of the PDR network genes mainly through Pdr3, and cooperatively drive Pdr3-dependent expression; while in response to ketoconazole SWI/SNF mediates Pdr1-dependent-drug-induced expression of the PDR network genes in *S. cerevisiae* (Fig. 6h).

**Roles of *C. glabrata* Rtt106 and SWI/SNF in antifungal resistance.** We next examined the roles of Rtt106 and SWI/SNF for antifungal resistance in *C. glabrata*, a pathogenic fungus closely related to *S. cerevisiae*. A major difference between *C. glabrata* and *S. cerevisiae* in the PDR network regulation is that *C. glabrata* has the single PDRE-binding transcription factor called CgPdr1 which shares homology almost equally with the *S. cerevisiae* Pdr1 and Pdr3 proteins, and possesses a blend of the properties of each[50]. Interestingly, loss of CgRtt106 (CAGL0H08041g) caused reduced induction of mRNA of the multidrug transporter genes *CgCDR1*, *CgCDR2* and of other PDR network genes, in cells transiently treated with ketoconazole (Fig. 7a orange bars and Supplementary Fig. 8a). An even more striking reduction was observed in the absence of SWI/SNF subunit CgSnf2 (Fig. 7a grey bars and Supplementary Fig. 8a). In contrast, in YPD without antifungal drugs, loss of CgRtt106 or CgSnf2 did not consistently reduce mRNA of the PDR network genes (Fig. 7a and Supplementary Fig. 8b). These results suggest that CgRtt106 and CgSWI/SNF drive the expression of the PDR network genes in *C. glabrata*, although their contributions appear limited to drug-induced expression.

We next tested if CgRtt106 and CgSWI/SNF localise at the *CgCDR1* promoter (which contains multiple PDRE and PDRE-like sequences; Fig. 7b), and found they bind at least two sites within the *CgCDR1* promoter (Fig. 7c, d, proximal and distal). Ketoconazole treatment did not increase CgRtt106 or CgSWI/SNF binding to the *CgCDR1* promoter (Fig. 7c, d), suggesting that these factors are already bound to the *CgCDR1* promoter before induction of its expression by the drug. CgRtt106 and CgSWI/SNF also localise at the *CgCDR1* promoter in YPD (Supplementary Fig. 8c), with higher background binding of CgRtt106 to a control locus probably because of its function in DNA replication.

We finally observed that along with a hyper-sensitive *Cgcdr1Δ*, *Cgsnf2Δ* and, to a lesser extent, *Cgrtt106Δ* exhibited increased sensitivity to ketoconazole and fluconazole (Fig. 7e). Taken together, these results demonstrate that CgRtt106 and CgSWI/SNF upregulate the expression of multidrug transporter genes and confer antifungal resistance in *C. glabrata*. CgRtt106 and CgSWI/SNF are therefore potential therapeutic targets to combat antifungal resistance of *C. glabrata*.

**Discussion**
Regulation of the PDR network genes is central to the fungal drug response, but is incompletely understood. In this study, we found that Rtt106 and SWI/SNF upregulate PDR network gene expression and confer drug resistance in *S. cerevisiae* and *C. glabrata*. In *S. cerevisiae*, Rtt106 is recruited to the promoter of the multidrug transporter gene *PDR5* exclusively through the transcription factor Pdr3, which drives Pdr3-mediated expression of *PDR5*, and confers resistance to azole antifungals. Furthermore, we showed that SWI/SNF plays a crucial role in both Pdr1- and Pdr3-mediated expression of the PDR network genes and confers azole drug resistance. Finally, we observed that CgRtt106 and CgSWI/SNF are important for the expression of the PDR network genes in response to ketoconazole in *C. glabrata*, and therefore for its resistance to azole antifungals, highlighting those factors as potential therapeutic targets to combat antifungal resistance of *C. glabrata*.

Our results and a published report[20] suggest that SWI/SNF regulates the expression of *PDR5* by shifting the TSS-associated +1 nucleosome, which is critical for transcription initiation. How does Rtt106 upregulate the expression of *PDR5*? One possible explanation is that Rtt106 recruits SWI/SNF to the *PDR5* promoter to activate its expression, similar to the histone promoters[33]. However, while the +1 nucleosome at the *PDR5* promoter was shifted in the *snf2* mutant, in the absence of Rtt106 nucleosome positioning was not obviously changed[29] (Supplementary Fig. 7c), suggesting that the effect of Rtt106 on *PDR5* transcription is not directly mediated through the SWI/SNF effect on nucleosome position. We instead favour a model where Rtt106 regulates 'turnover' of the nucleosomes surrounding the NDR at the *PDR5* promoter, to enable transient TATA-box region exposure allowing transcriptional regulators to access their binding sites. Although histone exchange at promoters is not simply related to transcriptional activity[21,51], we suspect that at the *PDR5* promoter nucleosome turnover may depend on Rtt106 and affect transcription. Interestingly, in *S. cerevisiae* rapid histone turnover also has been shown to occur at other sites where Rtt106 is enriched, including *tRNA* and *snoRNA*[51] (Fig. 2e and Supplementary Fig. 1d).

Proteins regulating the recruitment of Rtt106 to histone promoters (i.e. Hir1, Asf1, Rtt109 and Yta7) also contribute to its recruitment to the *PDR5* promoter. How these proteins contribute to Rtt106 recruitment remains to be elucidated. They might be required to mediate the physical interaction between Rtt106 and Pdr3, which would be consistent with our observation that purified Pdr3 and Rtt106 did not interact (Supplementary Fig. 4f). Another possibility is that these factors contribute to Rtt106 recruitment indirectly, e.g. through regulating the binding of Pdr3 to the *PDR5* promoter and/or regulating the abundance of Pdr3 (Supplementary Fig. 4d, e).

Do other histone chaperones have a role at the promoters? It has been reported that histone chaperone Asf1 mediates nucleosome disassembly from the *PHO5* promoter for its activation in yeast[22,23]. In human, histone chaperones HIRA and ASF1a localise at active promoters where BRG1, a human SWI/SNF complex subunit, also localises[52]. Interestingly, at promoters,

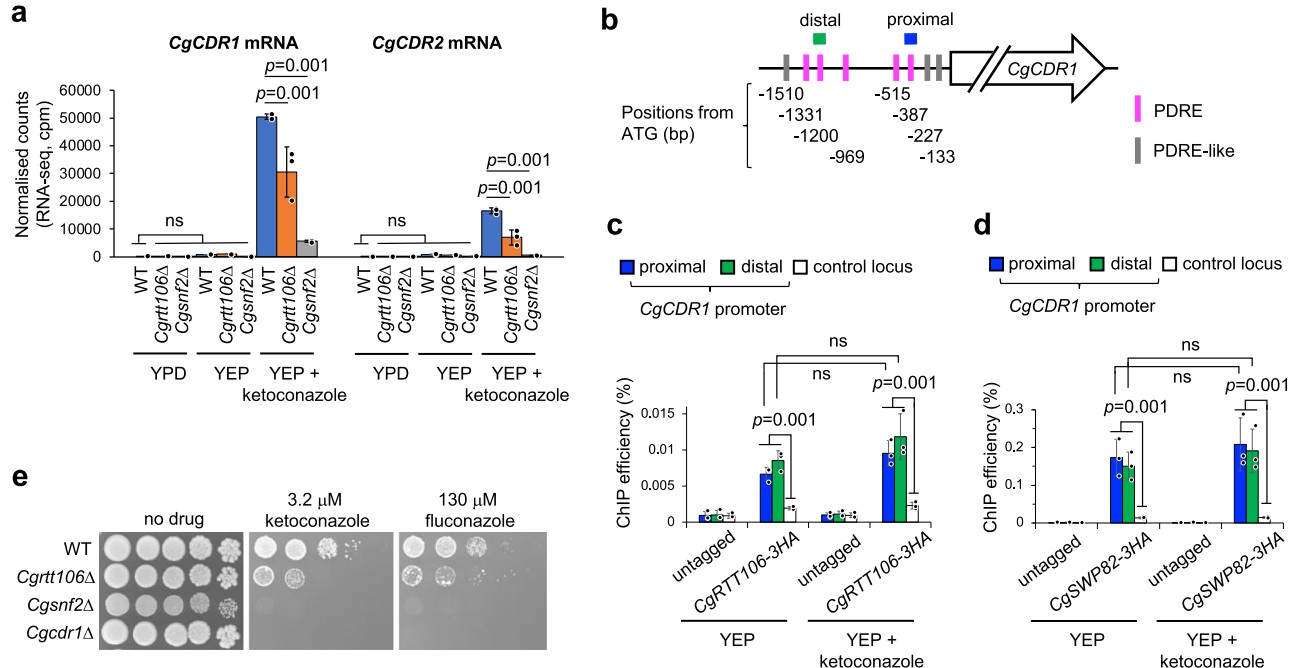

**Fig. 7 CgRtt106 and CgSWI/SNF mediate antifungal resistance by upregulating expression of the multidrug transporter genes in *C. glabrata*. a** RNA-seq analysis reveals reduction of mRNA of multidrug transporter genes *CgCDR1* and *CgCDR2* in the absence of CgRtt106 and CgSnf2 upon transient ketoconazole treatment in YEP. Data represents mean of three biological replicates. Significance determined by one-way ANOVA with post-hoc Tukey HSD test. ns, $P > 0.05$. **b** Positions of PDRE and PDRE-like motif in the *CgCDR1* promoter. **c, d** ChIP-qPCR analyses of CgRtt106 (**c**) and the CgSwp82 SWI/SNF subunit (**d**) at the *CgCDR1* promoter in YEP with and without ketoconazole. Proximal (blue) and distal (green) PDRE sites (illustrated in **b**) were analysed. Control locus (white) is coding region of the *CgADY3* gene. Data represent mean of three biological replicates. Significance determined by one-way ANOVA with post-hoc Tukey HSD test. ns, $P > 0.05$. **e** Sensitivity to azole antifungal drugs of the *C. glabrata* reference strain (ATCC 2001) and *C. glabrata* lacking CgRtt106, CgSnf2 or CgCdr1. Five-fold dilutions of cell culture were spotted on YPD or YPD containing ketoconazole or fluconazole, and incubated at 30 °C for 2–3 days.

human HIRA and ASF1a essentially localise to nucleosome-free TSSs[52], reminiscent of our findings that yeast Rtt106 is enriched at wider nucleosome-free regions upstream of TSSs (Fig. 2g). HIRA mediates the replacement of canonical H3 with the H3.3 variant at promoters[52], and continuous H3.3 exchange has been proposed to keep these sites transiently exposed, allowing transcription factors to access their binding sites[53]. These findings suggest that histone chaperones have a conserved role at promoters in yeast and human.

CgRtt106 regulates drug-induced expression of the PDR network genes in *C. glabrata*, although in *S. cerevisiae* ScRtt106 does not. We suspect this difference reflects the fact that CgPdr1 possesses a blend of ScPdr1 and ScPdr3 properties, so that CgRtt106 binds CgPdr1 (like ScRtt106 binds ScPdr3) and regulates CgPdr1-mediated, drug-induced expression. We detected no obvious contribution of CgRtt106 nor CgSWI/SNF to the basal expression of the PDR network genes in *C. glabrata*, which might reflect the fact that without drugs PDR gene expression is very low, compared to expression upon ketoconazole treatment (Fig. 7a, YPD vs YEP + ketoconazole). This difference is less marked in *S. cerevisiae*. How CgRtt106 and CgSWI/SNF upregulate drug-induced expression of the PDR network genes remains to be elucidated. GOF mutations in CgPdr1 make it hyperactive, resulting in high multidrug transporter expression and thus driving antifungal resistance in clinical isolates of *C. glabrata*[9,54]. It will be of great interest to test whether loss of CgRtt106 or CgSWI/SNF re-sensitises clinical isolates of *C. glabrata* that have developed antifungal resistance to azole antifungal drugs. SWI/SNF and Rtt106 might also contribute to drug resistance and virulence in other *Candida* species. *C. albicans* SWI/SNF mediates overexpression of the drug efflux protein

Mdr1 due to GOF mutations in the transcription factor Mrr1, thus conferring fluconazole resistance[55]. *C. albicans* Rtt106 and *C. albicans* SWI/SNF affect hyphae formation[56,57], which indirectly influences drug tolerance and pathogenicity.

Our findings in this study suggest that Rtt106 and SWI/SNF are general regulators of the PDR network genes and confer drug resistance in *S. cerevisiae*, and their roles are largely conserved in the pathogenic fungus *C. glabrata*. Interfering with the binding of CgRtt106 and CgSWI/SNF to CgPdr1 would be a strategy to prevent overexpression of the multidrug transporter genes in drug-resistant clinical isolates of *C. glabrata*. In this context, a small molecule inhibitor disrupting the interaction between CgPdr1 and CgMediator has already been successfully used to re-sensitise drug-resistant clinical isolates to azole antifungals[58]. Since Rtt106 and some SWI/SNF subunits are fungal-specific, these factors offer attractive therapeutic targets to combat antifungal drug resistance in *C. glabrata* and potentially other *Candida* species.

## Methods

**Yeast strains and plasmids**. *S. cerevisiae* strains and *C. glabrata* strains used are listed in Supplementary Table 3. Oligonucleotides used are listed in Supplementary Data 7. Epitope-tagging and gene disruptions were carried out using standard PCR-based gene insertion methods[59]. The hyperactive *pdr1-3* mutant in the S288C (BY4741) background was constructed by replacing *pdr1Δ::natNT2* in the genome of strain VNY70 with a PCR product containing the entire *pdr1-3* locus amplified from the original *pdr1-3* strain (congenic to W303-1A)[15] following DNA break induction at natNT2 using CRISPR-Cas9[60]. Hyperactive *pdr3-2* in the S288C (BY4741) background was constructed by the same approach, but with *pdr3Δ::natMX* (VNY45) and the original *pdr3-2* mutant[15]. Sequencing of the inserted *pdr1-3* and *pdr3-2* genes in the S288C backgrounds confirmed that *pdr1-3* contains the following mutations compared to the S288C reference genome: Q411K, F815S, T820A, T921I and loss of 5Ns at 1011 aa, while *pdr3-2* contains Q56R, H110Q, G834S, A885T, N916S and E941G. Strains for minichromosome

isolation were constructed from the SILAC compatible strain SHY201[61] by replacing the *TRP1* gene with the fragment containing LacI-3FLAG and *URA3* (amplified from pRS406-CMV-LacI-3FLAG[46] by PCR), followed by transformation with minichromosome plasmids pUC-TALO8 or pUC-TALO8-PDR5pr. Plasmid pUC-TALO8-PDR5pr (PDR5pro) was constructed by inserting the fragment of the *PDR5* promoter (from −997 to −241 bp upstream of ATG, not including the TATA-box) into the NdeI site of plasmid pUC-TALO8 (Empty)[46], using NEBuilder HiFi Assembly Master Mix (E2621L, NEB). The inserted sequences were confirmed by sequencing. For luciferase reporter assay, YCplac22-based plasmid containing the luciferase gene fused to the *PDR5* promoter (from −1000 bp to −1 bp upstream of ATG) was used. *E. coli* expression vectors, pGEX-6P-1-ScPDR1, pGEX-6P-1-ScPDR3, pGEX-6P-1-ScRTT106, and pET28a-ScRTT106 were constructed using NEBuilder HiFi Assembly Master Mix (E2621L, NEB).

**ChIP-sequencing and ChIP-qPCR analyses.** Cells were fixed with 1% formaldehyde at room temperature for 30 min and then at 4 °C overnight with gently shaking. The fixed cells were washed in ice-cold PBS three times and used for ChIP. ChIP was performed essentially as described[62] using mouse monoclonal anti-HA antibody (5 μg, HA.11 clone 16B12, Covance or Bioglegend) or mouse monoclonal anti-myc antibody (5 μg, M047-3, MBL, purchased from Caltag Medsystems) with modifications: cells were disrupted using a FastPreP-24 bead beater (MP Biomedicals), and sonication of DNA was performed using a Bioruptor (Diagnode). For ChIP-seq analysis, libraries of input and ChIP DNA were generated by NEBNext Ultra II DNA Library Prep Kit for Illumina (E7645L, NEB) and sequenced on NextSeq 500 to generate single-end 75 bp reads. For ChIP-qPCR analysis, input and ChIP DNA were analysed by LightCycler 480 II (Roche) using Light cycler SYBR Green master reagent (4707516001, Roche) or Takyon No ROX SYBR MasterMix (UF-NSMT-B0701, Eurogentic). ChIP efficiency (ChIP/Input) at each locus was calculated as the mean of three biological replicates, each one of which has three technical replicates.

**Bioinformatic analysis of ChIP-sequencing data.** The Galaxy web interface (https://usegalaxy.org) was used for bioinformatics analysis of ChIP-seq data. Sequence reads from fastq files were mapped against the latest version of the yeast genome sacCer3 using BWA (version 0.7.17.4). Non-uniquely mapped reads were removed by filtering out all reads with mapping quality less than 20 using SAMtools Filter SAM or BAM. To view read coverage distribution across the genome, BAM files generated with BWA were first converted into bigWig format using DeepTools bamCoverage (version 3.1.2.0.0), with the following settings: Bin size, 25 bp; Scaling/Normalisation method, normalise coverage to 1×; and Extend reads to the given average fragment size, 300 bp. BigWig datasets of Input and ChIP DNA were then displayed separately using Integrated Genome Browser (IGV, version 2.4.16).

To calculate Rtt106 enrichment or occupancy at each promoter region, enrichment of sequence reads in ChIP samples over corresponding Input samples was first calculated using DeepTools bamCompare (version 3.1.2.0.0), with bin size 50 bp. To extract values around the TSS of each gene, heatmap was then plotted using annotation of TSS[63], DeepTools computeMatrix (version 3.1.2.0.0) and plotHeatmap (version 3.1.2.0.1). Finally, the data table underlying the plots was exported and utilised for calculating Rtt106 enrichment in a region spanning 500 bp upstream of the TSS of each gene. To calculate the fold change of Rtt106 signals in *pdr1Δ pdr3Δ*, *hir1Δ* or *yta7Δ* over WT at each promoter, enrichment of ChIP sequence reads in those mutants over WT was calculated.

In clustering analysis, promoters bound by Rtt106 were classified into nine clusters based on Rtt106 enrichment at promoters in WT and its change in *pdr1Δ pdr3Δ*, *hir1Δ* and *yta7Δ*. The thresholds were determined to classify PDRE-containing promoters into cluster 1. All the promoters bound by Rtt106 were inspected manually to find features within promoters.

**Treatment of yeast cultures with azole antifungal drugs.** Cells were treated with azole antifungal drugs as described[7] with minor modifications. Briefly, cells grown in YPD (1% yeast extract, 2% peptone, 2% glucose) at 30 °C overnight until stationary phase were pelleted and washed with sterilised MilliQ purified water twice, resuspended in YEP (1% yeast extract, 2% peptone) to an optical density (OD$_{600}$) of 0.6, and further incubated for 24 h at 30 °C. Cells were treated with 40 μM ketoconazole (K1003, Merck) or 130 μM fluconazole (PHR1160, Merck) for 15 min for RNA preparation and ChIP-qPCR. For sampling cells in the glucose-starved condition (YEP), cells were collected before the addition of azole antifungal drugs to the cultures.

**RNA sequencing.** Total RNA was prepared from cells using Macherey-Nagel NucleoSpin RNA Columns (11922402, Fisher Scientific) according to the manufacturer's protocol. ERCC (External RNA Controls Consortium) spike-in controls were added to the RNA samples to be used as internal controls. Libraries were prepared with the Illumina TruSeq Stranded mRNA kit and sequenced using the High Output 1 × 75 kit on the Illumina NextSeq 500 platform producing 75 bp single-end reads.

**Bioinformatic analysis of RNA sequencing data.** Raw reads were filtered using TrimGalore! (version 0.6.4) with a phred quality score threshold of 30. ERCC spike-in reads were removed by alignment to the ERCC reference genome using HISAT2 (version 2.1.0)[64]. The cleaned reads were aligned to the *S. cerevisiae* reference genome using HISAT2 (version 2.1.0)[64]. SAMtools (version 1.9)[65] was used to process the alignments and reads were counted at gene locations using featureCounts (part of the sub read version 1.6.2 package)[66] utilising the parameter to split multi-mapped reads as a fraction across all genes that they align to and the parameter for stranded analysis. edgeR (version 3.26.8)[67] was used normalise reads counts, retaining genes that had a CPM (count per million) value of more than one in three or more samples, and to detect which of the genes had a significant differential change in expression (FDR < 0.05). Gene Ontology Term Finder (https://www.yeastgenome.org/goTermFinder, version 0.86) was used for gene ontology analysis.

**Northern blot analysis.** Northern blot analysis was performed as described[68]. Briefly, total RNA was separated on the HT (HEPES-Triethanolamine) gel and then transferred onto Hybond N + membrane (GERPN203B, Merck) by capillary transfer. RNA was crosslinked with the membrane by UV irradiation. *PDR5* mRNA and a loading control *ARF1* mRNA[69] were detected using biotin-labelled oligo DNA probes (Merck) and North2South Chemiluminescent Hybridisation and Detection Kit (17097, Thermo Scientific) according to the manufacturer's protocol. rRNAs were detected as loading controls, by SYBR Gold staining (S11494, Thermo Scientific). The images were captured using ChemiDoc system (Bio-Rad).

**Luciferase reporter assay.** Cells carrying luciferase reporter plasmids were grown in SC-Trp (Kaiser synthetic complete media lacking tryptophan (Formedium), supplemented with 2% glucose) at 30 °C until log phase and harvested. Cells were washed with sterilised MilliQ purified water, and resuspended in Lysis buffer A (100 mM potassium phosphate buffer, pH 7.8, 0.2% Triton X-100). Cells were disrupted using a FastPreP-24 bead beater (MP Biomedicals). The cleared lysate was prepared by centrifugation at 15,000 × *g* for 10 min at 4 °C. Luciferase activity was measured using Bright-Glo Luciferase Assay System (E2610, Promega) and Omega Plate Reader (BMG Labtech) and calculated as relative luminescence units per mg of total soluble protein.

**Total protein preparation.** To detect Pdr5-3HA, total protein was prepared from cells by the TCA method as follows. Cells grown in 7 ml of YPD at 30 °C until log phase were washed in ice-cold water, then resuspended in 150 μl of ice-cold YEX lysis buffer (1.85 M NaOH, 7.5% 2-mercaptoethanol) and incubated on ice for 10 min on ice. After adding 150 μl of ice-cold 50% (w/v) TCA to the cell suspension, the samples were incubated on ice for 10 min, and then span for 5 min at 10,000 × *g* at 4 °C to precipitate proteins. The precipitated proteins were resuspended in the protein sample buffer (40 mM Tris-HCl, pH 6.8, 8 M Urea, 5% SDS, 0.1 mM EDTA, 1% 2-mercaptothanol) and incubated at 37 °C for 15 min. Total proteins for detecting epitope-tagged Pdr3 were prepared by alkaline method: cells were washed once with water, incubated in 0.1 M NaOH for 5 min at room temperature, spun down, resuspended in SDS sample buffer (1610737, Bio-Rad) and incubated at 85 °C for 5 min. Total proteins prepared were separated by SDS-PAGE using 4–20% TGX stain-free gel (4568095, Bio-Rad), and transferred onto the PVDF membrane using the Trans-Blot Turbo Transfer system (Bio-Rad). Tagged proteins were detected using anti-HA antibody (1:5000, Covance or Biolegend, HA.11, clone 16B12), anti-FLAG M2 antibody (1:5000, Sigma or Merck, F1804, clone M2) or anti-myc antibody (1:50,000, abcam, ab9106) and Clarity Western ECL substrate (1705061, Bio-Rad). As a loading control, total protein was detected using Stain-Free technology (Bio-Rad).

**Purification of the SWI/SNF complex.** The SWI/SNF complex was purified from a protease-deficient yeast strain containing 3× Flag tag at the C-terminus of Snf6 (Snf6-3Flag). The tagged yeast strain was grown in 2 L of YPD at 30 °C. Cells were collected by centrifugation and washed with 40 ml of cold PBS. Cells were resuspended in 15 ml of Lysis buffer B (20 mM HEPES-NaOH, pH 7.4, 350 ml NaCl, 10% glycerol, 0.1% Tween 20, 1 mM PMSF, Halt protease and phosphatase inhibitor (11844111, Fisher Scientific)) and lysed in a FastPreP-24 bead beater (MP Biomedicals). Cell debris was removed by centrifugation at 16,000 × *g* at 4 °C for 20 min. The cleared lysate was mixed with anti-FLAG M2 antibody (30 μg, F1804, Merck) bound to Dynabeads Protein G (10765583, Fisher Scientific) and incubated at 4 °C for 2 h. The beads were washed five times in 1 ml of Lysis buffer B. Bound proteins were eluted with 200 μl of Lysis buffer B containing 1 mg/ml 3× FLAG peptide (Merck) by incubating at room temperature for 30 min, with shaking. The elution step was repeated twice, yielding 400 μl of eluate. The SWI/SNF complex was further purified and concentrated using Amicon Ultra centrifugal filter unit (MWCO 100 kDa, UFC5100, Merck), removing smaller proteins including 3× FLAG peptide and exchanging buffer to PBS containing 0.02% Tween 20 and 50% glycerol. The purified proteins were analysed by 4–15% TGX gel (4568085, Bio-Rad) followed by SYPRO Ruby staining (S12000, Thermo Fisher Scientific).

**Purification of His-tagged Rtt106.** His-Rtt106 was purified from *E. coli* using Ni-NTA resin (30210, QIAGEN). *E. coli* cells NiCo21(DE3) (C2529H, NEB) carrying

pET28a-ScRTT106 were grown in 100 ml LB-kanamycin at 37 °C to OD$_{600}$ 0.6. Expression of His-Rtt106 was induced by incubating with 0.4 mM IPTG at 37 °C for 3 h. Cells were then collected by centrifugation, resuspended in 2 ml of Lysis buffer C (50 mM NaH$_2$PO$_4$, pH 8.0, 1 M NaCl, 10 mM imidazole, 5 mM 2-mercaptoethanol, 0.1% Triton X-100, 10% glycerol, 1 mM PMSF, 3 μg/ml pepstatin A and Halt protease inhibitor), and lysed by sonicating on ice (Soniprep 150), with four cycles of 10 sec on then 10 sec off, at an amplitude of 8 μm. Cell debris was removed by centrifugation at 16,000 × g at 4 °C for 20 min. The cleared lysate was mixed with 250 μl of Ni-NTA resin and incubated at 4 °C for 60 min. After washing the resin with Wash buffer (50 mM NaH$_2$PO$_4$, pH 8.0, 1 M NaCl, 20 mM imidazole, 5 mM 2-mercaptoethanol, 0.1% Triton X-100, 10% glycerol, 1 mM PMSF, 3 μg/ml pepstatin A), His-Rtt106 was eluted with Elution buffer (50 mM NaH$_2$PO$_4$, pH 8.0, 300 mM NaCl, 250 mM imidazole, 5 mM 2-mercaptoethanol, 0.1% Triton X-100, 10% glycerol).

**GST pull-down assay**. E. coli cells NiCo21(DE3) carrying pGEX-6P-1, pGEX-6P-1-ScPDR1, pGEX-6P-1-ScPDR3, and pGEX-6P-1-ScRTT106 were grown in LB-ampicillin at 30 °C until OD$_{600}$ reaches 0.5. The expression of GST and GST-fusion proteins was induced by incubating with 0.1 mM IPTG at 16 °C for 18 h. Cells were then collected by centrifugation, resuspended in cold Lysis buffer D (PBS containing 1 M NaCl, 0.1% Triton X-100, 1 mM DTT, 1 mM PMSF, Halt protease inhibitor), and lysed by sonicating on ice (Soniprep 150), with four cycles of 10 sec on, 10 sec off, at an amplitude of 8 μm. Cell debris was removed by centrifugation at 16,000 × g at 4 °C for 20 min. The cleared lysate was mixed with 20 μl of glutathione-Sepharose beads (GE17-0756-01, Merck) and incubated at 4 °C for 30 min. After washing four times with Lysis buffer D, the beads bound to GST or GST fusions were incubated with purified SWI/SNF, purified His-Rtt106 or yeast cell lysate containing HA-tagged Rtt106 in Binding buffer (PBS, 0.1% Triton X-100, 1 mM DTT, 1 mM MgCl$_2$, 10% glycerol, 1 mM PMSF, Halt protease inhibitor) at 4 °C for 2 h. After washing four times with binding buffer, bound proteins were eluted in SDS sample buffer, and analysed by SDS-PAGE and western blot. GST and GST fusions were detected using Stain-Free technology (Bio-Rad) or by Amido black staining (A8181, Merck). Snf6-3FLAG, His-Rtt106 and Rtt106-6HA were detected using anti-FLAG M2-HRP (1:5000, A8592, Merck), anti-His (1:5000, 652502, Biolegend), and anti-HA (1:5000, HA.11 clone 16B12, Biolegend) antibodies, respectively.

**Spot assay**. Yeast cultures grown in YPD at 30 °C overnight were diluted and then spotted on YPD plates containing fluconazole (PHR1160, Merck) or ketoconazole (K1003, Merck). The plates were then incubated at 30 °C for 2–4 days.

**Minichromosome isolation**. Minichromosomes were isolated essentially as described[46]. Cells carrying minichromosome plasmids were grown in 1 L of SILAC media[70] (Kaiser SC-Arg-Lys-Trp (Formedium), supplemented with 20 mg/L heavy arginine-HCl (Arg10, CK Isotopes) and 30 mg/L lysine-2HCl (Lys6, CK Isotopes) or, for a control, 20 mg/L light arginine-HCl and 30 mg/L light lysine-2HCl) at 30 °C until log phase and harvested. After washing cells in ice-cold water supplemented with 2 mM PMSF, cells were resuspended in buffer H150 (25 mM HEPES-KOH, pH 7.5, 150 mM KCl, 2 mM MgCl$_2$, 10% glycerol, 0.02% NP40, 1 mM PMSF, cOmplete Protease Inhibitor cocktail without EDTA (11873580001, Merck), PhosSTOP (4906845001, Merck), 5 mM nicotinamide) and disrupted using a FastPrep-24 bead beater (MP Biomedicals). Clarified lysates were prepared by centrifugation at 20,000 × g for 20 min three times. lacO-containing minichromosomes were isolated by immunoprecipitating LacI-3FLAG with anti-FLAG M2 antibody (12 μg, F1804, Merck) crosslinked to Dynabeads protein G (10765583, Fisher Scientific) by dimethyl pimelimidate. Proteins on purified minichromosomes were eluted in Elution buffer (50 mM ammonium bicarbonate, 0.1% Rapi-Gest SF (186001860, Waters), 1 mM IPTG). The eluted proteins were analysed by SYPRO Ruby staining and western blot analysis.

**Mass spectrometry**. Peptides digested by trypsin in solution (n = 1 biological sample) were analysed using an Orbitrap Q Exactive Plus mass spectrometer (Thermo Scientific) equipped with a Dionex U3000 RSLCnano liquid chromatography system configured for pre-concentration on C18 PepMap 100 (300 μm i.d. × 5 mm) then peptide separation on EASY-Spray PepMap RSLC C18 (75 μm i.d. × 50 cm) over a 245 min elution gradient as described previously[71] with minor modifications. Two solvents were formed, solvent A was made from water/formic acid (1000:0.1) and solvent B from water/acetonitrile/formic acid (20:80:0.1). An increasing proportion of solvent B was used to separate peptides along a gradient: 3–40% from 5 to 205 min; 40–90% from 205 to 206 min; 90% from 206 to 216 min; 90–3% from 216 to 217 min, followed by re-equilibration of the column (3% Solvent B from 217–245 min). Mass spectra were acquired using a Top 10 data-dependent method starting at 25 min and lasting 205 min. Full MS scans were conducted between 375 and 1750 m/z at a resolution 70,000 (m/z 200), automatic gain control 3E + 6 and a maximum injection time 50 ms. Following each survey scan the 10 most intense ions of charge state 2–5 were sequentially selected (isolation window 1.6 m/z) and fragmented in the higher-energy collisional dissociation (HCD) cell at a normalized collision energy of 28%. MS/MS scans were conducted at a resolution 17,500, automatic gain control 5E + 4 and a maximum

injection time 100 ms. Dynamic exclusion of 40 s prevented ions from triggering subsequent data-dependent scans during that time.

**Mass spectrometry data analysis**. RAW data obtained from SILAC samples were processed with MaxQuant (version 1.6.5.0)[72] using the standard settings for Orbitrap against a S. cerevisiae protein database (UP000002311_559292). Carbamidomethylation of cysteines was set as a fixed modification and oxidation of methionines, deamidation of asparagines and glutamines, and protein N-terminal acetylation as variable modifications. Minimal peptide length was set to seven amino acids and a maximum of two missed Trypsin/P cleavages was allowed. SILAC quantification was performed using standard settings and a Label min ratio count of 5. RAW data obtained from the SWI/SNF-enriched fraction was processed with Proteome Discoverer (version 2.2.0.388, Thermo Fisher Scientific) incorporating Mascot Server (version 2.5) using the standard settings for Orbitrap, against a S. cerevisiae protein database (UP000002311_559292). Carbamidomethylation of cysteines was set as a fixed modification and oxidation of methionines, and phosphorylation of serines, threonines and tyrosines as variable modifications. A maximum of three missed Trypsin cleavages was allowed. A strict target FDR of 0.01 for the decoy database search was used.

**Statistics and reproducibility**. Statistical analysis was performed by two-tailed unpaired Student's t-test, one-way ANOVA with post-hoc Tukey HSD test, or two-sided Mann–Whitney test as described in each figure legend. For Northern blot and GST pull-down analyses and testing protein level of Pdr5 (Fig. 4b), each experiment was repeated three times independently with similar results, and one representative data of the replicates are presented. The experiments for confirming the expression of a series of epitope-tagged Pdr3 (Supplementary Fig. 3b, c) were performed once with at least two different isolates which showed similar results. Minichromosome isolation and purification of the SWI/SNF complex followed by SYPRO Ruby staining (Supplementary Fig. 6a, b) were repeated three times independently with similar results; data from one representative experiment are shown. Mass spectrometry analysis of isolated minichromosome was performed once and therefore proteins quantified at least 5 ratio counts were presented, allowing reliable quantification of proteins in the sample analysed. The binding of SWI/SNF to the PDR5 promoter (found by the minichromosome isolation experiment) was subsequently confirmed by ChIP-qPCR analysis with three biological replicates with similar results. Enrichment of the SWI/SNF complex by immunopurification was assessed by comparing the band pattern on the gel with that in published data[48] (n = 3 biological replicates, with similar results), and subsequently confirmed by mass spectrometry (n = 1).

**Reporting summary**. Further information on research design is available in the Nature Research Reporting Summary linked to this article.

## Data availability
The data that support this study are available from the corresponding author upon reasonable request. All raw-data files for ChIP-seq and RNA-seq data were uploaded to ArrayExpress under accession numbers: E-MTAB-9787 and E-MTAB-11013. The mass spectrometry proteomics data have been deposited to the ProteomeXchange Consortium via the PRIDE partner repository with the dataset identifier PXD028798. Publicly available datasets used in this study are GSM3452517 (Swi3 ChEC-seq), GSM3177770 (MNase-seq, +Snf2), GSM3177771 (MNase-seq, -Snf2), GSM3452556 (TBP ChIP-seq, +Snf2), GSM3452557 (TBP ChIP-seq, -Snf2), SRX648019 (MNase-seq, WT), SRX648516 (MNase-seq, rtt106Δ), SRX648532 (Yta7 ChIP-seq), GSM1565066 (TSS-seq), Saccharomyces cerevisiae genome (sacCer3, April 2011) (https://hgdownload.soe.ucsc.edu/goldenPath/sacCer3/chromosomes/) and Saccharomyces cerevisiae protein database (UP000002311_559292). Source data are provided with this paper.

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

## Acknowledgements

We thank Karl Kuchler, Carol Munro, Donna MacCallum, Delma Childers and Ian Stansfield for plasmids and strains; Ben Rutter, Madeleine Murray and Tom Jayasekara for help with material constructions and homology search; Vamsi Gali, Anne Donaldson, Yuki Katou and Katsuhiko Shirahige for preliminary ChIP-seq data; Shin-ichiro Hiraga for MaxQuant analysis; Stefan Hoppler for access for equipment; Sophie Shaw and Antonio Ribeiro for bioinformatic analysis of the RNA-seq data and also data uploading to ArrayExpress and Anne Donaldson and Alexander Lorenz for careful reading of the manuscript. We are grateful to the core facilities at University of Aberdeen: the CGEBM facility for help with ChIP-seq and RNA-seq (Ewan Campbell and Zeynab Heidari), the Proteomics facility for mass spectrometry (David Stead) and the qPCR facility. Work was supported by Medical Research Council (MRC) Career Development Fellowship L019698/1 to T.K., and Wellcome Trust Strategic Award for Medical Mycology and Fungal Immunology 097377/Z/11/Z to T.K. and D.M.

## Author contributions

Conceptualisation: V.N. and T.K.; Formal analysis: V.N. and T.K.; Investigation: V.N., D.M. and T.K.; Writing—original draft: T.K.

## Competing interests

The authors declare no competing interests.
