## [Peer Review File · Nature Communications]

REVIEWER COMMENTS

Reviewer #1 (Remarks to the Author):

The manuscript by Nikolov et al. presents a very clearly written description of a technically sound mechanistic investigation of the role of the histone chaperone Rtt106 in the expression of the PDR1/3 regulon in *S. cerevisiae*. The PDR1/3 regulon is of particular importance in fungi since it has been shown that activation of this regulon is critical for innate and acquired (through gain of function mutations) azole resistance in clinical isolates. This mainly happens through the transcriptional activation of efflux pumps similar to *S. cerevisiae* PDR5 that pump the drug and out of the cell. Rtt106 has been previously implicated in the deposition of histones in chromatin and the regulation of the *S. cerevisiae* histone genes, however this is the first report, to the best of my knowledge, of a specific involvement in the expression of the PDR1/3 regulon

The central observation upon which this manuscript is that Rtt106 occupancy is enriched at genes that have a PDR element (PDRE) in their promoter and that the expression of these PDRE genes is impacted under conditions in which they are not induced by a gain of function mutation. Interestingly, the upregulation of the PDR1/3 regulon resulting from treatment with ketoconazole is not strongly impacted by Rtt106 deletion. However, this seems to be consistent with their finding that Rtt106 primarily works with the transcription factor Pdr3, while the response to induction by ketoconazole is primarily orchestrated through Pdr1. Despite this small impact on activation by ketoconazole, the absence of Rtt106 does make the yeast somewhat more sensitive to ketoconazole in a resistance assay. The Rtt106 deletion has a much higher impact on resistance to fluconazole, which does not activate Pdr1. It had been previously established by work in the Moye-Rowley lab that mitochondrial dysfunction also upregulated the PDR1/3 regulon and led to increased azole resistance. In an interesting follow up to these findings, the authors find that the deletion Rtt106 completely abrogates the increased azole resistance exhibited by the strain with the defective mitochondria. The last major part of the paper is connecting the PDRE and the Swi/Snf chromatin remodeling complex by virtue of a chromatin pull-down and mass spec experiment on the PDR5 promoter. When finding enrichment of Swi/Snf subunits, the authors began to look at the role of the complex and found that it was extremely important for the expression of the PDR1/3 regulon and for azole resistance. A quick check on the human fungal pathogen, *C. glabrata*, showed that deletion of Rtt106 and of Snf2 (the catalytic subunit of Swi/Snf) had similar increased sensitivity to ketoconazole as they had observed in *S. cerevisiae*. On the positive side, this is a clear, impactful and well performed study that establishes some new relationships between a histone chaperone and drug resistance that may be relevant to mechanisms of resistance in more distantly related fungal pathogens to their model system in baker's yeast. On the negative side the dependence on Pdr3 basal expression on Rtt106 rather than the dependence of Pdr1 induced expression makes the paper slightly less impactful since many of the acquired resistance mechanisms in *C. albicans* and other common pathogens rely primarily on activation of the Pdr1/3 ortholog, usually through a gain of function mutation. My concerns about the paper mainly focus on places where more caution has to be used in interpreting the results, given the limitations of the experiments as currently presented. These concerns could be addressed by qualifying the claims, or doing further experiments. Below I describe my specific concerns as well as some more technical opportunities for improving the manuscript below:

Major Points

- Although the ChIP data clearly show that the recruitment of Rtt106, Swi/Snf and Pdr1/3 to the PDRE are correlated, there is no evidence presented that these interactions are

direct. In many places a direct interaction is implied and in Figure 8 a direct interaction is explicitly portrayed. Either some biochemical data or other in vivo technique (FRET?) would be required to demonstrate direct interactions, which would strengthen the mechanistic component of the paper.

- Lines 308-311: It is suggested that Asf1 and Rtt109 contribute to the recruitment of Rtt106 to the PDR5 promoter. It is important to establish whether or not this is through Pdr3 or not, to establish the relative contribution of different factors. A ChIP of Pdr3 in the deletion strains would address this issue.
- Line 323-325 - The experiment showing that a GOF mutation in Pdr3 bypasses the Rtt16 dependence of ketoconazole resistance is a very important experiment due to GOF mutations being a major mechanism of acquired azole resistance. I propose this experiment should be moved out of the supplementary data and into the main body of the paper.
- Line 380-381 - While I agree that there looks to be a small shift in the nucleosome peak in the Snf2 null strain it is important to establish whether this is statistically significant or not. Is there a metric that the authors can apply to this data to establish statistical significance?
- Line 382-383 - I don't think that you can read too much into the reduced TBP levels at the Snf2 mutant as support for reduced accessibility of the TATA box. TBP occupancy almost always scales with transcription levels, so the fact that reduced TBP levels accompany reduced expression of PDR5 really doesn't tell us much about the mechanism.
- Figure 4F - It is important to comment on the considerable difference in ketoconazole resistance between the two *fzo1 rtt106* double null strains. Moreover, given the focus of the paper on the expression of PDR5 and the rest of the Pdr1/3 regulon, I was surprised not to see any data on the expression of these genes in the *fzo1 rtt106* double null strain since this would strongly impact the general conclusions of the paper.

Technical Points

- On line 51-53 it is mentioned that Pdr1/3 can be activated by direct binding to ketoconazole. I think it is important to qualify this statement since the only study to show this was done in *C. glabrata*.
- Line 109-110. Unless there is some data that show that the class of proteins represented by Rtt106 and Swi/Snf could be targeted by small molecule therapeutics, I think it is wishful thinking that they could actually be realistic "potential therapeutic targets."
- Line 230-233. The authors state that they believe the increased expression of some genes resulting from the loss of Rtt106 may be a result of reduced nucleosome occupancy. Is the reduced occupancy observed? Does Rtt106 even ChIP to these regions?

Reviewer #2 (Remarks to the Author):

This manuscript implicates the histone chaperone Rtt106 and the chromatin remodeling complex (represented here by its Snf2 subunit) as important contributors to expression of PDR gene expression in *S. cerevisiae* and its pathogenic relative, *Candida glabrata*. A large amount of data are provided that are argued to link Rtt106 promoter recruitment with its association with the Pdr3 transcription factor. ChIP-seq of Rtt106 found that some PDR

genes were enriched with binding of this histone chaperone, especially the ABC transporter gene PDR5. Not all PDR genes were observed to be bound by Rtt106 and loss of Rtt106 caused a reduction in expression of roughly 20 PDR genes. RNA measurements using *pdr1*, *pdr3* or *pdr3 rtt106* null strains were carried out and argued to indicate that Pdr3/Rtt106 were important in basal expression while Pdr1 was required for drug-induced transcription. A somewhat perplexing result was that, although Pdr1 is necessary for drug induction, only Pdr3 and Rtt106 are required for high level resistance to azole drugs. Loss of Rtt106 also blocked the *rho0* induction of Pdr3 but had no effect on the high level azole resistance driven by hyperactive alleles of either PDR1 or PDR3. A fragment of the PDR5 promoter was used to purify proteins that bound to this regulatory element and several transcription factors were identified to be enriched by this approach. These included Pdr1, Pdr3, Mediator components, Rtt106 and the chromatin remodeling Swi/Snf complex. Recruitment of this complex was evaluated using Snf2 as a representative subunit. Loss of Snf2 reduced PDR5 expression and drug resistance. Snf2 recruitment required Pdr3, Rtt106 and Pdr1. The effect of Snf2 was argued to be positioning of the +1 nucleosome which is important in transcriptional initiation. Loss of RTT106 or SNF2 also reduced fluconazole resistance in the pathogenic yeast *Candida glabrata*.

Clearly, there is a large amount of work provided here. It is interesting that Rtt106 is enriched on some PDR promoters and this seems to be Pdr3-dependent. The authors do a nice job of integrating information from other labs concerning chromatin structure into their data. However, there are some major issues with aligning their model with what is already known from the literature concerning PDR gene regulation in *S. cerevisiae*. Their views of the relative contributions of Pdr1 and Pdr3 to gene regulation are different from what has been published with no justification as to why their revisitation is the more accurate picture. A second and even more significant issue comes from a lack of consideration of the global importance of chromatin structure influencing gene expression. Certainly, histone positioning and remodeling is a crucial issue in transcription and its regulation. Rtt106 and Snf2 will absolutely impact many different genes and showing that the PDR loci are among these does not establish a specific role for these factors.

Specific concerns are listed below.

1. Evidence from two different groups defined PDR1 as the major contributor to PDR5 expression in *S. cerevisiae* (Mahé et al Mol Micro 1996 20:107-117, Katzmann, et al MCB 1994 14:4653-4661). Pdr3 expression is very poor unless the autoregulatory circuit of the corresponding gene is engaged (as in *rho0* cells). Expression of Pdr1 is roughly 7 times that of Pdr3 under non-induced conditions (taken from SGD website). If the authors provided data that Pdr3 had a higher specific activity as a transcriptional regulator, their arguments would be stronger. In its present form, this seems like a revision of the literature that is not well-supported and more driven by a need to fit with these new data.

2. Along these same lines, here is a statement from the legend to Figure 8: "Pdr1 is not required for expression of the PDR network genes although it may make a minor contribution to SWI/SNF recruitment." This is simply incorrect. Experiments in these same papers above and many others show that loss of PDR1 causes decreased PDR5 expression in log phase growth. The issue here is a lack of careful consideration of the large body of older data raises concerns about the manner in which these new findings are coherently integrated.

3. The drug induction experiments are done under conditions of glucose starvation and catabolite derepression while all other drug experiments employ standard culture conditions. The authors are clear in pointing out that this has been used before in the literature but this raises doubts about what is being measured here. The drug induction

shows the expected genetic dependence but is happening under completely different metabolic conditions under which all the other assays were performed. Either the other drug resistance measurements should be performed on YEP media or the induction experiments should be done using standard growth conditions.

4. The roles of Rtt106 and SWI/SNF are investigated fairly strictly within their roles in PDR gene expression and drug resistance. These proteins are certain to have global roles in regulation of expression. Controls should be supplied showing that loss of these regulators have effects that are specific to PDR genes. If stresses not handled by the PDR pathway (oxidative, metal, etc) are tested, are similar effects on expression seen. The possibility that the effects described here are seen at many different genes, rather than the presented tight focus on the PDR pathway, should be addressed. If the impact of Rtt106 and SWI/SNF can be explained by global chromatin structural defects, then the advances provided by this work would be harder to identify.

5. The mechanism proposed to explain the effect of Rtt106 on PDR5 gene expression involves repositioning of a histone around the +1 position of transcription. This is thought to occlude the TATA box and reduce access to this key region of the PDR5 promoter. The data provided are not very strong and lack comparisons with other promoters that would strengthen the conclusions. Only RPB2 is provided as a comparison promoter. What about the other PDR promoters that don't respond to Rtt106? The authors should provide these comparisons as well as to accurately determine TFIID binding to this region via ChIP and qPCR. These graphic depictions showing binding do not allow quantitative comparisons to be made across different promoters.

6. The observation that RTT genes are involved in both chromatin structure and azole resistance in *C. glabrata* have been observed in earlier work (PLoS Pathog. 2012 Aug; 8(8): e1002863). The group of Kaur only analyzed RTT107/109 but these proteins are part of the same regulatory pathway as Rtt106. This earlier work did not explicitly examine defects in chromatin assembly but did show under conditions where these RTT genes are critical, chromatin structure was disturbed.

Reviewer #3 (Remarks to the Author):

The manuscript by Nikolov et al. describes compelling data indicating that the histone chaperone Rtt106 and the SWI/SNF chromatin remodeling complex play key roles in contributing to the basal gene expression of the PDR network in *S. cerevisiae*. The studies are well done and comprehensive, and provide interesting insights into chromatin regulation of drug resistance genes. They also reveal that this might be conserved in the pathogenic yeast *C. glabrata*, although the supporting data are very limited.

The paper would in principle be suitable for Cell Reports if the authors could expand the studies in *C. glabrata* to increase the clinical relevance of their findings, given the clear roles of the PDR pathway in this species in azole resistance (e.g., Caudle et al. Eukaryot Cell 2011; Nishikawa et al. Nature 2016). In particular, it would be interesting to see whether the *C. glabrata* Pdr1 transcription factor binds to *C. glabrata* SWI/SNF and/or Rtt106 and recruit them to target genes (e.g. CDR1/2), and what the ChIP-seq profiles of Rtt106 and SWI/SNF look like in the presence and absence of azoles, and whether their recruitment to the CDR genes depend on Pdr1 in this species. Also, what do the RNA-seq profiles look like in *C. glabrata* strains lacking Rtt106 and SWI/SNF components +/- azoles?

We thank the Reviewers for their thoughtful consideration of our findings and useful suggestions, which have improved the manuscript substantially. We have addressed the Reviewers' comments as described below:

Major changes:

1. In the revised manuscript, we now show that Rtt106 contributes to hyper-resistance of a Pdr3 gain-of function (GOF) mutant to an azole antifungal

In the original manuscript, we used mainly *S. cerevisiae* strains made in the S288C background (one of the most widely used backgrounds particularly for genome-wide studies). Exceptions were the hyper-resistant strains (*pdr1-GOF* and *pdr3-GOF*), for which the W303-1A was used. To be consistent, in the revised manuscript we have constructed *pdr1-GOF* and *pdr3-GOF* mutations in the S288C background and now include analysis of these strains. Interestingly, we now see that Rtt106 mediates *PDR5* overexpression in *pdr3-GOF* in the S288C background, and loss of Rtt106 sensitised *pdr3-GOF* mutant to the azole ketoconazole. Those new data (Northern blot analysis and spot assay) have been added to Fig. 4 (g and h). The main text was modified accordingly (page 13). Differences in genetic backgrounds are also discussed on page 10-11.

SWI/SNF is also critical for *PDR5* overexpression caused by *pdr3-GOF*, and for *pdr3-GOF*-mediated drug resistance in the S288C background. Interestingly, combining *pdr1-GOF* and deletion of *SNF6* caused a growth defect even without azole drugs (through an as yet unknown mechanism). Those new data are added to Fig. 5 (i-n), and the main text modified accordingly (page 15). These new data emphasise the important roles of Rtt106 and SWI/SNF in hyper-resistance to antifungals caused by GOF mutations, which are closely related to hyper-resistance in clinical isolates of *C. glabrata*.

2. Results related to drug resistance induced by mitochondrial dysfunction have been removed from the main figures because of reproducibility issues

Although a previous report (Hallstrom et al. 2000. JBC, 275: 37347–37356) shows that mitochondrial dysfunction causes hyper-resistance to azole antifungals, we did not observe such phenotype reproducibly in the strains we used in our study. We had two approaches to induce loss of mitochondrial DNA: deleting the *FZO1* gene important for mitochondrial outer membrane fusion, and treating cells with ethidium bromide (new Supplementary Figs. 3h and 3i), but only one out of 14 isolates show slightly increased resistance to ketoconazole (Supplementary Fig. 3i, isolate 7). We suspect that the single drug-resistant clone emerged as a chance outcome of genome instability caused by loss of mitochondrial DNA (Veatch et al. 2009. Cell, 137:1247-58). We suspect that different genetic backgrounds used in the previous report and our study may explain why we did not observe hyper-resistance of cells with mitochondrial dysfunction. We used S288C, while Hallstrom et al. used SEY6210, a strain constructed by Prof Scott Emr used in studies of autophagy (see Saccharomyces Genome Database Wiki https://wiki.yeastgenome.org/index.php/Commonly_used_strains). We prefer to use S288C consistently since the relative contributions of Pdr1 and Pdr3 to gene regulation in S288C seem to differ from those in SEY6210 and also W303-1A (see also our response to Reviewer 2, point 1). Because S288C does not reliably show the effect, we were not able to test if Rtt106 and SWI/SNF mediate drug resistance caused by mitochondrial dysfunction in this strain background.

Therefore, the original figures related to mitochondrial dysfunction-induced drug resistance have been removed because the phenotype was not observed reproducibly, and instead, new Supplementary figures were now added (Supplementary Figs. 3h and 3i) demonstrating that mitochondrial dysfunction does not generally cause increased drug resistance in the S288C genetic background. The main text was modified accordingly (page 13). These changes do not weaken the importance of this study, since we now show that both Rtt106 and SWI/SNF contribute to azole

hyper-resistance of a GOF mutant of Pdr3, an effect that is more directly related to the hyper-resistance observed in clinical isolates of *C. glabrata* (see Major point 1).

Point-by-point response:

Reviewer #1

The manuscript by Nikolov et al. presents a very clearly written description of a technically sound mechanistic investigation of the role of the histone chaperone Rtt106 in the expression of the PDR1/3 regulon in *S. cerevisiae*. The PDR1/3 regulon is of particular importance in fungi since it has been shown that activation of this regulon is critical for innate and acquired (through gain of function mutations) azole resistance in clinical isolates. This mainly happens through the transcriptional activation of efflux pumps similar to *S. cerevisiae* PDR5 that pump the drug and out of the cell. Rtt106 has been previously implicated in the deposition of histones in chromatin and the regulation of the *S. cerevisiae* histone genes, however this is the first report, to the best of my knowledge, of a specific involvement in the expression of the PDR1/3 regulon

The central observation upon which this manuscript is that Rtt106 occupancy is enriched at genes that have a PDR element (PDRE) in their promoter and that the expression of these PDRE genes is impacted under conditions in which they are not induced by a gain of function mutation. Interestingly, the upregulation of the PDR1/3 regulon resulting from treatment with ketoconazole is not strongly impacted by Rtt106 deletion. However, this seems to be consistent with their finding that Rtt106 primarily works with the transcription factor Pdr3, while the response to induction by ketoconazole is primarily orchestrated through Pdr1. Despite this small impact on activation by ketoconazole, the absence of Rtt106 does make the yeast somewhat more sensitive to ketoconazole in a resistance assay. The Rtt106 deletion has a much higher impact on resistance to fluconazole, which does not activate Pdr1. It had been previously established by work in the Moye-Rowley lab that mitochondrial dysfunction also upregulated the PDR1/3 regulon and led to increased azole resistance. In an interesting follow up to these findings, the authors find that the deletion Rtt106 completely abrogates the increased azole resistance exhibited by the strain with the defective mitochondria. The last major part of the paper is connecting the PDRE and the Swi/Snf chromatin remodeling complex by virtue of a chromatin pull-down and mass spec experiment on the PDR5 promoter. When finding enrichment of Swi/Snf subunits, the authors began to look at the role of the complex and found that it was extremely important for the expression of the PDR1/3 regulon and for azole resistance. A quick check on the human fungal pathogen, *C. glabrata*, showed that deletion of Rtt106 and of Snf2 (the catalytic subunit of Swi/Snf) had similar increased sensitivity to ketoconazole as they had observed in *S. cerevisiae*. On the positive side, this is a clear, impactful and well performed study that establishes some new relationships between a histone chaperone and drug resistance that may be relevant to mechanisms of resistance in more distantly related fungal pathogens to their model system in baker's yeast. On the negative side the dependence on Pdr3 basal expression on Rtt106 rather than the dependence of Pdr1 induced expression makes the paper slightly less impactful since many of the acquired resistance mechanisms in *C. albicans* and other common pathogens rely primarily on activation of the Pdr1/3 ortholog, usually through a gain of function mutation. My concerns about the paper mainly focus on places where more caution has to be used in interpreting the results, given the limitations of the experiments as currently presented. These concerns could be addressed by qualifying the claims, or doing further experiments. Below I describe my specific concerns as well as some more technical opportunities for improving the manuscript below:

1. Although the ChIP data clearly show that the recruitment of Rtt106, Swi/Snf and Pdr1/3 to the PDRE are correlated, there is no evidence presented that these interactions are direct. In many

places a direct interaction is implied and in Figure 8 a direct interaction is explicitly portrayed. Either some biochemical data or other in vivo technique (FRET?) would be required to demonstrate direct interactions, which would strengthen the mechanistic component of the paper.

We have added new figures showing that:

- Rtt106 binds Pdr3, but not directly
- SWI/SNF binds Pdr1 and Pdr3, likely through direct interactions
- SWI/SNF binds Rtt106, likely through a direct interaction

To test the physical interaction between Rtt106 and Pdr3, we performed GST pull-down assays using purified proteins or yeast cell lysate. GST-Pdr3 that was purified from *E. coli* expression system bound Rtt106 when we used yeast cell lysate containing HA-tagged Rtt106, but did not bind His-tagged Rtt106 purified from *E. coli* expression system. These results suggest that Rtt106 can bind physically to Pdr3, but there is another requirement for their binding, e.g., other proteins or post-translational modifications. New figures show these experiments (Fig. 4e and Supplementary Fig. 3g), as explained in the main text (page 13).

To test the interaction of SWI/SNF with Rtt106, Pdr1, and Pdr3, we first purified the SWI/SNF complex from yeast cells by immunoprecipitating Flag-tagged Snf6, a subunit of SWI/SNF. We observed that purified SWI/SNF bound GST-Rtt106, GST-Pdr1, and GST-Pdr3, suggesting that SWI/SNF physically interacts with them. Since the SWI/SNF-enriched fraction contains other proteins, e.g., the Mec1 checkpoint kinase, we cannot necessarily conclude that the interaction is 'direct' and we avoid stating that in the manuscript. Interestingly, peptides from Pdr1 were identified by mass spectrometry in the purified SWI/SNF fraction, providing further supporting evidence for interaction of SWI/SNF with Pdr1. These results are shown in Figs. 5f, 5g and 6d, and Supplementary Figs. 4d and 4e, and explained in the main text (pages 14-15 and 16).

2. Lines 308-311: It is suggested that Asf1 and Rtt109 contribute to the recruitment of Rtt106 to the PDR5 promoter. It is important to establish whether or not this is through Pdr3 or not, to establish the relative contribution of different factors. A ChIP of Pdr3 in the deletion strains would address this issue.

We attempted to test if Asf1 and Rtt109 affect binding of Pdr3 to the *PDR5* promoter. However the N-terminally Flag-tagged Pdr3 that we constructed was not functional (the strain was sensitive to ketoconazole). Therefore we could not address this question.

Instead, we now discuss the possibility that Asf1 and Rtt109 affect binding of Pdr3 to the *PDR5* promoter (Discussion on page 20).

We also added a bracket to the model figure (now Fig. 6j in the revised manuscript) to indicate the possibility that Asf1 and Rtt109 affect binding of Pdr3 to the *PDR5* promoter.

3. Line 323-325 - The experiment showing that a GOF mutation in Pdr3 bypasses the Rtt106 dependence of ketoconazole resistance is a very important experiment due to GOF mutations being a major mechanism of acquired azole resistance. I propose this experiment should be moved out of the supplementary data and into the main body of the paper.

New data demonstrating this effect for GOF mutations are now shown in the main Figures (Fig. 4h). See 'Major change, point 1' above.

4. Line 380-381 - While I agree that there looks to be a small shift in the nucleosome peak in the Snf2 null strain it is important to establish whether this is statistically significant or not. Is there a metric that the authors can apply to this data to establish statistical significance?

We have analysed the nucleosome occupancy change caused by loss of Snf2 at promoters in the published dataset (Kubik et al. 2019), and find that the +1 nucleosomes are shifted significantly at the PDR network gene promoters in the absence of Snf2, compared to promoters not bound by SWI/SNF. These new data and statistical tests have been added to Fig. 6 (g and h), and the main text modified accordingly (pages 16-17).

5 Line 382-383 - I don't think that you can read too much into the reduced TBP levels at the Snf2 mutant as support for reduced accessibility of the TATA box. TBP occupancy almost always scales with transcription levels, so the fact that reduced TBP levels accompany reduced expression of PDR5 really doesn't tell us much about the mechanism.

To avoid overinterpretation, we now used the milder expression "...consistent with the idea that..." in the main text (page 17).

Utilising the published dataset (Kubik et al. 2019), we have analysed TBP binding change caused by loss of Snf2 at promoters, and observe that TBP binding decreased in the absence of Snf2 at the PDR network gene promoters, compared to promoters not bound by SWI/SNF (Fig. 6i and the main text on page 17).

6. Figure 4F - It is important to comment on the considerable difference in ketoconazole resistance between the two *fzo1 rtt106* double null strains. Moreover, given the focus of the paper on the expression of PDR5 and the rest of the Pdr1/3 regulon, I was surprised not to see any data on the expression of these genes in the *fzo1 rtt106* double null strain since this would strongly impact the general conclusions of the paper.

Please see the 'Major change, points 2' section above.

7. On line 51-53 it is mentioned that Pdr1/3 can be activated by direct binding to ketoconazole. I think it is important to qualify this statement since the only study to show this was done in *C. glabrata*.

Activation of *S. cerevisiae* Pdr1 by direct binding to ketoconazole has been shown in Thakur et al. 2008 Nature. They showed that both ScPdr1 and ScPdr3 can bind ketoconazole directly. Furthermore, ScPdr1 was activated by direct binding to ketoconazole. Whether ScPdr3 is activated by binding to ketoconazole was however not shown in the Thakur et al. paper. The main text has been modified accordingly (page 3).

8. Line 109-110. Unless there is some data that show that the class of proteins represented by Rtt106 and Swi/Snf could be targeted by small molecule therapeutics, I think it is wishful thinking that they could actually be realistic "potential therapeutic targets."

We think that Rtt106 and SWI/SNF are potential therapeutic targets based on the previous findings related to Pdr1 and Mediator. Mediator, a large complex like SWI/SNF, interacts with Pdr1, and the interaction between Mediator and Pdr1 can be disrupted by a small molecule inhibitor called iKIX (Nishikawa et al. Nature 2016). iKIX can re-sensitise drug-resistant *C. glabrata* to azole antifungals. Given that example we think that small molecule therapeutics may work, dependent on how Rtt106 and SWI/SNF bind to Pdr1/3, making them potential therapeutic targets. We now discuss this issue with mention of the small molecule inhibitor for the Pdr1-Mediator interaction (page 21).

9. Line 230-233. The authors state that they believe the increased expression of some genes resulting from the loss of Rtt106 may be a result of reduced nucleosome occupancy. Is the reduced occupancy observed? Does Rtt106 even CHIP to these regions?

We have not observed a correlation of genes whose expression increased in *rtt106Δ* with Rtt106 binding at those genes.

Our discussion draws on previous studies of effects of *rtt106Δ* on nucleosome arrangement. Reduced histone deposition was observed at transcribing genes and at DNA replication origins in *rtt106Δ* (Imbeault et al. 2008 JBC; Zunder et al. 2012 PNAS). Also, increased nucleosome spacing was observed in *rtt106Δ* globally, particularly evident at highly expressing genes (Lombardi et al. 2015 Genetics). We think this is simply because nucleosome positions at highly expressing genes are well defined, as refinement of nucleosome position is coupled with transcription. Conversely, nucleosomes at genes with low expression were less well positioned across a cell population because of low transcription-mediated refinement of nucleosome positions (i.e., well-positioned 'peaks' of nucleosome occupancy are not visible at genes with low expression). Therefore, we could not address if nucleosome occupancy and spacing at genes with low expression are changed in the absence of Rtt106.

In our ChIP-seq analyses, Rtt106 was not enriched at genes with low expression. But, it is likely that Rtt106 is travelling with (or following) the DNA replication machinery everywhere along chromosomes, so that Rtt106 deposits histones at genes with low expression as it tracks through these genes coupled to replication. We suspect that nucleosomes at genes with low expression levels may be reduced in *rtt106Δ* through defects in this mechanism.

People may wonder why loss of Rtt106 appears to increase mRNA levels for most genes that normally show low expression (Fig. 3a). We suspect that actually, some level of unregulated transcription initiation occurs at many or most genes in *rtt106Δ* (because of reduced nucleosome occupancy or increased nucleosome spacing). Such a moderate effect can be expected to cause a more noticeable increase (i.e., greater fold change) of mRNA normally present at low levels, than of mRNA already present in high abundance.

We now cite the appropriate papers and have modified the main text to explain this argument (page 10).

Reviewer #2

This manuscript implicates the histone chaperone Rtt106 and the chromatin remodeling complex (represented here by its Snf2 subunit) as important contributors to expression of PDR gene expression in *S. cerevisiae* and its pathogenic relative, *Candida glabrata*. A large amount of data are provided that are argued to link Rtt106 promoter recruitment with its association with the Pdr3 transcription factor. ChIP-seq of Rtt106 found that some PDR genes were enriched with binding of this histone chaperone, especially the ABC transporter gene PDR5. Not all PDR genes were observed to be bound by Rtt106 and loss of Rtt106 caused a reduction in expression of roughly 20 PDR genes. RNA measurements using *pdr1*, *pdr3* or *pdr3 rtt106* null strains were carried out and argued to indicate that Pdr3/Rtt106 were important in basal expression while Pdr1 was required for drug-induced transcription. A somewhat perplexing result was that, although Pdr1 is necessary for drug induction, only Pdr3 and Rtt106 are required for high level resistance to azole drugs. Loss of Rtt106 also blocked the rho0 induction of Pdr3 but had no effect on the high level azole resistance driven by hyperactive alleles of either PDR1 or PDR3. A fragment of the PDR5 promoter was used to purify

proteins that bound to this regulatory element and several transcription factors were identified to be enriched by this approach. These included Pdr1, Pdr3, Mediator components, Rtt106 and the chromatin remodeling Swi/Snf complex. Recruitment of this complex was evaluated using Snf2 as a representative subunit. Loss of Snf2 reduced PDR5 expression and drug resistance. Snf2 recruitment required Pdr3, Rtt106 and Pdr1. The effect of Snf2 was argued to be positioning of the +1 nucleosome which is important in transcriptional initiation. Loss of Rtt106 or SNF2 also reduced fluconazole resistance in the pathogenic yeast *Candida glabrata*.

Clearly, there is a large amount of work provided here. It is interesting that Rtt106 is enriched on some PDR promoters and this seems to be Pdr3-dependent. The authors do a nice job of integrating information from other labs concerning chromatin structure into their data. However, there are some major issues with aligning their model with what is already known from the literature concerning PDR gene regulation in *S. cerevisiae*. Their views of the relative contributions of Pdr1 and Pdr3 to gene regulation are different from what has been published with no justification as to why their revisitation is the more accurate picture. A second and even more significant issue comes from a lack of consideration of the global importance of chromatin structure influencing gene expression. Certainly, histone positioning and remodeling is a crucial issue in transcription and its regulation. Rtt106 and Snf2 will absolutely impact many different genes and showing that the PDR loci are among these does not establish a specific role for these factors.

Specific concerns are listed below.

1. Evidence from two different groups defined PDR1 as the major contributor to PDR5 expression in *S. cerevisiae* (Mahé et al Mol Micro 1996 20:107-117, Katzmann, et al MCB 1994 14:4653-4661). Pdr3 expression is very poor unless the autoregulatory circuit of the corresponding gene is engaged (as in rho0 cells). Expression of Pdr1 is roughly 7 times that of Pdr3 under non-induced conditions (taken from SGD website). If the authors provided data that Pdr3 had a higher specific activity as a transcriptional regulator, their arguments would be stronger. In its present form, this seems like a revision of the literature that is not well-supported and more driven by a need to fit with these new data.

First, we would like to emphasise that there are already inconsistencies in the literature, and that our finding about contributions of Pdr1 and Pdr3 to gene regulation is consistent with previous report from Kemmeren et al. 2014. However, we do notice that our results are inconsistent with some other reports, and apologise for not discussing such differences in the original manuscript. We suspect such differences reflect differences on genetic backgrounds used.

Interestingly, it seems that the relative contributions of Pdr1 and Pdr3 to gene regulation are different dependent on genetic backgrounds tested, as summarised below.

- Mahé et al. 1996 used strains congenic to W303-1A, and showed Pdr1 as the main factor involved.
- Katzmann et al. 1994 used SEY6210, and showed that Pdr1 and Pdr3 equally contribute to gene regulation (note that they did not see Pdr1 as the main factor).
- Kemmeren et al. 2014 used S288C, and found Pdr3 to be the main factor involved. In a large-scale microarray analysis, they observed that *PDR5* mRNA was reduced in *pdr3Δ* to 38% of normal, but found no effect in *pdr1Δ*.

In our study, we used S288C (a strain used in the systematic sequencing project, and the strain background now used most widely, particularly for genome-wide analyses). Our finding is consistent with Kemmeren et al. 2014, which also used S288C.

Although these differences are interesting, elucidating the mechanism causing such difference is beyond the scope of this study. We now discuss these differences in the main text (pages 10-11).

Also please see the 'Major change' section above.

2. Along these same lines, here is a statement from the legend to Figure 8: "Pdr1 is not required for expression of the PDR network genes although it may make a minor contribution to SWI/SNF recruitment." This is simply incorrect. Experiments in these same papers above and many others show that loss of PDR1 causes decreased PDR5 expression in log phase growth. The issue here is a lack of careful consideration of the large body of older data raises concerns about the manner in which these new findings are coherently integrated.

The statement in the original Figure 8 was intended to describe the data being presented, in which deletion of *PDR1* did not cause reduction of *PDR5* mRNA in YPD in the strain background we used. We do agree that the statement does not describe with the observations in other strain backgrounds (as discussed in Point 1 above). The differences have now been discussed in the main text (pages 10-11), and we have removed the statement from the figure legend to avoid mis-interpretation (now, Fig. 6j).

3. The drug induction experiments are done under conditions of glucose starvation and catabolite derepression while all other drug experiments employ standard culture conditions. The authors are clear in pointing out that this has been used before in the literature but this raises doubts about what is being measured here. The drug induction shows the expected genetic dependence but is happening under completely different metabolic conditions under which all the other assays were performed. Either the other drug resistance measurements should be performed on YEP media or the induction experiments should be done using standard growth conditions.

We do accept the limitation Reviewer 2 pointed out here. But, it would not be possible to test drug resistance on YEP media, since without a carbon source cells cannot grow. Similarly, the induction experiments cannot be done using standard growth condition (i.e., YPD) as ketoconazole treatment does not induce expression of *PDR5* in YPD as shown in Supplementary Fig. 2c. We now clearly mention why we could not use YPD for the induction experiments (page 11).

The limitation is now mentioned on page 12 as follows, "*Note that although we showed that ketoconazole treatment does not induce PDR5 expression in YPD liquid media (Supplementary Fig. 2c), the level of PDR5 expression in cells grown long-term on YPD plate with drug is not known.*"

4. The roles of Rtt106 and SWI/SNF are investigated fairly strictly within their roles in PDR gene expression and drug resistance. These proteins are certain to have global roles in regulation of expression. Controls should be supplied showing that loss of these regulators have effects that are specific to PDR genes. If stresses not handled by the PDR pathway (oxidative, metal, etc) are tested, are similar effects on expression seen. The possibility that the effects described here are seen at many different genes, rather than the presented tight focus on the PDR pathway, should be addressed. If the impact of Rtt106 and SWI/SNF can be explained by global chromatin structural defects, then the advances provided by this work would be harder to identify.

We thank Reviewer 2 for highlighting this point to us.

First, our study does not propose that loss of Rtt106 and SWI/SNF has effects specific ONLY to the PDR genes. As it has been reported in many papers (e.g., Sudarsanam et al. 2000), SWI/SNF

regulates expression of a subset of genes, e.g., stress-response genes, NOT only PDR genes. We showed that loss of Rtt106 affects mRNA levels of genes with low expression, NOT only PDR genes.

We believe that Reviewer 2's main concern here is whether a global chromatin defect in the absence of Rtt106 and SWI/SNF affects *PDR5* mRNA levels 'indirectly'.

A direct role of SWI/SNF at the PDR network genes is clearer in our revised manuscript, since we have added data showing modulation of the TSS-associated +1 nucleosome by SWI/SNF at the PDR network genes (Figs. 6g and 6h, also see Reviewer 1 Point 4 above, and also Point 5 below).

Several results in this study support the idea that increased *PDR5* mRNA in *rtt106Δ* is not simply caused by global chromatin defect. Important in this respect is our finding that loss of Rtt106 reduced *PDR5* overexpression in *pdr3-GOF*, but not in *pdr1-GOF* (new Fig. 4g). This selective effect suggests a direct regulation of Pdr3 by Rtt106, rather than a consequence of global chromatin defect. This point is now mentioned in the main text (page 13).

5. The mechanism proposed to explain the effect of Rtt106 on *PDR5* gene expression involves repositioning of a histone around the +1 position of transcription. This is thought to occlude the TATA box and reduce access to this key region of the *PDR5* promoter. The data provided are not very strong and lack comparisons with other promoters that would strengthen the conclusions. Only RPB2 is provided as a comparison promoter. What about the other PDR promoters that don't respond to Rtt106? The authors should provide these comparisons as well as to accurately determine TFIID binding to this region via ChIP and qPCR. These graphic depictions showing binding do not allow quantitative comparisons to be made across different promoters.

Just to clarify first, we proposed SWI/SNF (not Rtt106) repositions the +1 nucleosome at the *PDR5* promoter. We have now added new Figures (Figs. 6g and 6h) showing that +1 nucleosomes at the PDR network genes (not just at the *PDR5* promoter) are generally shifted in the absence of SWI/SNF, compared to promoters not bound by SWI/SNF. We also observed that in the absence of SWI/SNF, TBP binding decreased significantly at the PDR gene promoters, compared to promoters not bound by SWI/SNF (new Fig. 6i). Note that TBP ChIP-seq in Kubik et al. 2019 was performed in the presence of spike-in control, allowing quantitative comparisons. The main text has been modified accordingly (pages 16-17).

6. The observation that RTT genes are involved in both chromatin structure and azole resistance in *C. glabrata* have been observed in earlier work (PLoS Pathog. 2012 Aug; 8(8): e1002863). The group of Kaur only analyzed RTT107/109 but these proteins are part of the same regulatory pathway as Rtt106. This earlier work did not explicitly examine defects in chromatin assembly but did show under conditions where these RTT genes are critical, chromatin structure was disturbed.

We thank Reviewer 2 for drawing our attention to this interesting paper. It does not seem directly related to our current work, since we investigated direct roles of Rtt106 and SWI/SNF in drug resistance, rather than importance of general chromatin remodelling once *Candida* is engulfed by host immune cells. However given our long-term interest in the importance of these pathways, this paper will influence our next future works.

Reviewer #3:

The manuscript by Nikolov et al. describes compelling data indicating that the histone chaperone Rtt106 and the SWI/SNF chromatin remodeling complex play key roles in contributing to the basal

gene expression of the PDR network in *S. cerevisiae*. The studies are well done and comprehensive, and provide interesting insights into chromatin regulation of drug resistance genes. They also reveal that this might be conserved in the pathogenic yeast *C. glabrata*, although the supporting data are very limited.

The paper would in principle be suitable for Cell Reports if the authors could expand the studies in *C. glabrata* to increase the clinical relevance of their findings, given the clear roles of the PDR pathway in this species in azole resistance (e.g., Caudle et al. *Eukaryot Cell* 2011; Nishikawa et al. *Nature* 2016). In particular, it would be interesting to see whether the *C. glabrata* Pdr1 transcription factor binds to *C. glabrata* SWI/SNF and/or Rtt106 and recruit them to target genes (e.g. CDR1/2), and what the ChIP-seq profiles of Rtt106 and SWI/SNF look like in the presence and absence of azoles, and whether their recruitment to the CDR genes depend on Pdr1 in this species. Also, what do the RNA-seq profiles look like in *C. glabrata* strains lacking Rtt106 and SWI/SNF components +/- azoles?

To investigate roles of Rtt106 and SWI/SNF in regulating expression of the PDR network genes in *C. glabrata*, we have now performed RNA-seq and ChIP-qPCR analyses. Interestingly, loss of Rtt106 or Snf2 in *C. glabrata* caused reduction of mRNA of multidrug transporter genes *CgCDR1* and *CgCDR2*, and other PDR network genes, in ketoconazole-treated cells (new Fig. 7a and new Supplementary Fig. 6a). These results suggest that Rtt106 and SWI/SNF drive expression of the PDR network genes in response to ketoconazole in *C. glabrata*. In contrast, in the absence of azole antifungals, loss of Rtt106 and Snf2 did not change expression of the PDR network genes consistently (new Supplementary Fig. 6b). These effects probably reflect the fact that the single *C. glabrata* Pdr1 transcription factor possesses a blended function of *S. cerevisiae* Pdr1 and Pdr3. We have carried out ChIP-qPCR analyses of Rtt106 and SWI/SNF in *C. glabrata*, and now include data confirming that they localise at the *CgCDR1* promoter (Figs. 7b and 7c). Therefore our paper now clearly demonstrates roles for Rtt106 and SWI/SNF in regulating expression of the PDR network genes in *C. glabrata*, and the main text has been modified accordingly (pages 17-18).

REVIEWER COMMENTS

Reviewer #1 (Remarks to the Author):

The authors have done a careful and comprehensive job addressing the concerns with the initial submission. The results build a solid foundation for the future studies of the role of Rtt106 in antifungal resistance mechanisms in human fungal pathogens.

Reviewer #2 (Remarks to the Author):

In my opinion, the revision of this manuscript does not address the concerns I expressed in my first review. A highlighted piece of new data nearly lacks any controls and issues raised both by another reviewer and I were not addressed.

The new data consists in part of demonstration that recombinant GST-Pdr3 can bind to Rtt106 from yeast cells. This is done with a single control consisting of GST alone. How can it be concluded that this GST-Pdr3 is folded correctly in any way? Does this protein interact with Med15 as shown by the Naar lab? Then recombinant Rtt106 is shown to be unable to interact with GST-Pdr3 which is interpreted to implicate another protein or some post-translational modification. This is a serious overinterpretation of what may be concluded or really even suggested from such a negative experiment.

Both reviewer 1 and I mentioned the importance of demonstrating that the effect of Rtt106 on Pdr3 and PDR5 expression was direct. Reviewer 1 suggested a Pdr3 ChIP experiment which would have been an excellent addition to this work. The fact that both *rtt106* and *pdr3* null mutants cause the same effect on PDR5 could be due to loss of Rtt106 blocking either DNA-binding of Pdr3 or Pdr3 expression. The authors state they were unable to perform this experiment as the one epitope tag version of Pdr3 they constructed was nonfunctional. I can understand how this might be a problem in some organisms but this is *S. cerevisiae*. Nearly every gene has been tagged with TAP and/or GFP, not to mention that epitope-tagged versions of PDR3 have already been described in the literature. This was a disappointing choice and weakens confidence in the conclusions presented.

Again, I shared the concerns of reviewer 1 over the small shift in nucleosome occupancy over the PDR5 promoter caused by loss of Snf2. Additionally, Tbp recruitment to PDR5 was suggested to be impacted by Snf2. The response was to simply re-discuss their use of the data of others. I have no concerns with these other data but this misses the point raised. There may be a slight reduction in binding of either a nucleosome or Tbp but I fail to understand why one would rely on previously published work to validate this essential feature of this current study.

I also noted that the ketoconazole induction experiments are all done on media lacking a carbon source while all plate assays contain a carbon source. Nothing was done to address this beyond arguing this is the nutritional condition required to observe induction. These data (induction and drug resistance) are interpreted together yet the growth conditions in which they are observed are quite different. The absence of a carbon source would induce, at a minimum, catabolite derepression and certainly trigger a severe protein degradation phenotype as well as blocks to translation. I do not agree that these vastly different growth conditions may be interpreted together.

This manuscript is difficult to integrate into the more than 3 decades of study of the Pdr system as little effort is made to ensure that the current data actually are consistent with the past. Just dismissing these discrepancies as strain differences is not helpful especially when only a single strain is used here. S288c is the strain that was used for genomic sequencing but it is hardly wild-type. It contains a range of mutations including a defective HAP1 gene. Additionally, work exists using the *S. cerevisiae* knockout disruption mutant strain (BY4742) that is consistent with PDR1 being the key

driver of drug resistance (Fardeau, et al (2007) JBC 282: 5063). BY4742 is very closely related to the BY4741 strain used predominantly here. What is normally done when there is some unusual genetics (like Pdr3 being the dominant driver of PDR5 and drug resistance) as argued here, is to confirm this in other backgrounds. This is not done and isn't addressed since they authors can find one other paper that seems to agree with this.

Finally, I found it hard to get around the conclusion that a pdr1 null prevented azole induced gene expression yet was not as critical as PDR3 for resistance. Somehow this is explained by Pdr3 being important in basal expression. This seems to be an odd explanation since, at least to me, basal expression is defined as the level of expression of a gene in the absence of some regulating signal (like ketoconazole evidently is for PDR5).

These weaknesses led to my lack of enthusiasm for this work.

Reviewer #3 (Remarks to the Author):

The authors have addressed all comments to my satisfaction.

We are glad to know that Reviewers 1 and 3 are satisfied with our first revised manuscript.

Reviewer #1 (Remarks to the Author):

The authors have done a careful and comprehensive job addressing the concerns with the initial submission. The results build a solid foundation for the future studies of the role of Rtt106 in antifungal resistance mechanisms in human fungal pathogens.

Reviewer #3 (Remarks to the Author):

The authors have addressed all comments to my satisfaction.

We have further addressed the Reviewer 2's comments as described below:

Reviewer #2 (Remarks to the Author):

In my opinion, the revision of this manuscript does not address the concerns I expressed in my first review. A highlighted piece of new data nearly lacks any controls and issues raised both by another reviewer and I were not addressed.

The new data consists in part of demonstration that recombinant GST-Pdr3 can bind to Rtt106 from yeast cells. This is done with a single control consisting of GST alone. How can it be concluded that this GST-Pdr3 is folded correctly in any way? Does this protein interact with Med15 as shown by the Naar lab? Then recombinant Rtt106 is shown to be unable to interact with GST-Pdr3 which is interpreted to implicate another protein or some post-translational modification. This is a serious overinterpretation of what may be concluded or really even suggested from such a negative experiment.

We thank Reviewer 2 for pointing out this issue. However a GST pull-down assay with a single control consisting of GST alone is standard and acceptable. In our case, a positive control is especially unnecessary since GST-Pdr3 binds to Rtt106 when using yeast cell lysate with HA-Rtt106, providing reassurance that the GST-Pdr3 folds correctly. Nonetheless, to avoid overinterpretation, we now modified the text: "Rtt106 shows a higher affinity for GST-Pdr3 than GST alone, when testing cell lysate from yeast with HA-tagged Rtt106 (Fig. 4e), suggesting that Rtt106 binds Pdr3 in this condition." (see Page 13).

Reviewer 2 is moreover mistaken: Thakur et al. 2008 Nature (the Naar lab) did not show if GST-Pdr3 expressed in *E. coli* binds Med15, in fact what they showed was interaction between GST-Pdr1 and Med15.

Notably, Reviewer 1 who originally requested this experiment was satisfied completely with the new GST pull-down data presented in the first revision.

We do, however, accept the possibility that the recombinant Rtt106 (i.e., His-Rtt106) does not fold properly, a possibility that was not mentioned in the first revised manuscript. To avoid overinterpretation we now modified the text: "However, His-Rtt106 purified from *E. coli* expression system was not co-precipitated with GST-Pdr3 (Supplementary Fig. 4f), implying there may be another molecular requirement for binding, or possibly that recombinant Rtt106 is not properly folded to bind Pdr3 in the condition tested. How Rtt106 binds Pdr3 remains to be elucidated" (Page 13).

Both reviewer 1 and I mentioned the importance of demonstrating that the effect of Rtt106 on

Pdr3 and *PDR5* expression was direct. Reviewer 1 suggested a Pdr3 ChIP experiment which would have been an excellent addition to this work. The fact that both *rtt106* and *pdr3* null mutants cause the same effect on *PDR5* could be due to loss of Rtt106 blocking either DNA-binding of Pdr3 or Pdr3 expression. The authors state they were unable to perform this experiment as the one epitope tag version of Pdr3 they constructed was nonfunctional. I can understand how this might be a problem in some organisms but this is *S. cerevisiae*. Nearly every gene has been tagged with TAP and/or GFP, not to mention that epitope-tagged versions of *PDR3* have already been described in the literature. This was a disappointing choice and weakens confidence in the conclusions presented.

Just to clarify, in the first review comments, neither Reviewers 1 nor 2 mentioned the importance of demonstrating that the effect of Rtt106 on Pdr3 and *PDR5* expression was direct. Reviewer 1 just suggested that we should test by Pdr3 ChIP if contributions of Asf1 and Rtt109 to the recruitment of Rtt106 to the *PDR5* promoter are through Pdr3 or not. Reviewer 2 asked if Rtt106 and SWI/SNF have a global effect or a specific effect on the PDR pathway (addressed in the first revision).

Nonetheless, we have now made further efforts to construct a functional, epitope-tagged *PDR3* strain as follows:

First, we carried out a literature search to find an epitope-tag less likely to compromise Pdr3 function (and to avoid non-functional strains, e.g., 3FLAG-Pdr3 we constructed before).

Mamnun et al. 2002 Mol. Microbiol. and Delahodde et al. 2001 Mol. Microbiol. mentioned that N-terminally tagged Pdr3 with 2HA and GFP are functional, respectively, although actual evidence was not shown. In contrast, Akache et al. 2004 JBC mentioned that N-terminally tagged Pdr3 is not functional, and therefore used C-terminally tagged Pdr3 (again, no actual evidence shown). We therefore decided to test all combinations, i.e., FLAG, HA and Myc tag on N- or C-terminus of Pdr3.

As shown in new Supplementary Figure 4a-c, we found that C-terminally 13Myc-tagged *PDR3* is the best strain based on the assessment of expression and also the strain's resistance to ketoconazole. Unexpectedly, strains with N-terminally 2HA- or 3HA- tagged *PDR3* are abnormally sensitive to ketoconazole (to a similar extent as the *pdr3* deletion mutant). We think that this is due to low or no expression of the HA-Pdr3 proteins in our system. Also unexpectedly, C-terminally tagged *PDR3* with 6HA is hyper-resistant to ketoconazole, and therefore this is not an ideal strain for ChIP. So, we used the *PDR3-13Myc* strain for Pdr3 ChIP experiments to test its recruitment to the *PDR5* promoter, examining if it is affected by loss of Rtt106, Snf2, or Asf1 as follows.

By ChIP-qPCR analysis using Pdr3-13myc, we have found that Rtt106 is not essential for Pdr3 recruitment to the *PDR5* promoter (new Supplementary Figure 4d). However, we observed ~50% reduction of Pdr3 binding to the *PDR5* promoter in *rtt106Δ* compared to WT, which could be simply caused by reduction of protein and mRNA levels of *PDR3* in *rtt106Δ* (Supplementary Figs. 4d and 4e and Supplementary Table 3). While not ruling out some direct contribution of Rtt106 to Pdr3 recruitment, these results suggest clearly that Rtt106 regulates expression of *PDR3*, one of the PDRE-containing genes. Reduction of Pdr3 protein level and its binding to the *PDR5* promoter in *rtt106Δ* may be one of the causes of reduced *PDR5* expression in *rtt106Δ*. Our results still suggest that Rtt106 regulates *PDR5* expression by acting directly at the *PDR5* promoter (not only indirectly through regulating *PDR3* expression) as a ~50% reduction of Pdr3 binding at the *PDR5* promoter in *rtt106Δ* cannot alone explain why both *rtt106Δ* and *pdr3Δ* mutants cause the same effect on *PDR5* expression. We now modified the main text accordingly (Page 13).

Similar to Rtt106, it is likely that SWI/SNF contributes to expression of *PDR3*, *PDR5* and other PDR genes through directly acting at their promoters as well as through positive feedback of Pdr3 expression, in the absence of drugs. The main text was now modified “As expected, the protein level of Pdr3 in *snf2Δ* is reduced, compared to WT (Supplementary Fig. 4e), consistent with reduction of mRNA level of *PDR3* in *snf2Δ* (Supplementary Table 7). Concomitantly, Pdr3 binding to the *PDR5* promoter is reduced in *snf2Δ*, compared to WT (Supplementary Fig. 4d). Taken together, these results and the evidence of SWI/SNF binding to the *PDR5* promoter suggest that SWI/SNF regulates *PDR5* expression by acting directly at the *PDR5* promoter, as well as through regulating gene expression of the transcription factor *PDR3*.” See Page 16.

Asf1 might affect Rtt106 recruitment to the *PDR5* promoter indirectly through Pdr3, since Pdr3 binding to the *PDR5* promoter is reduced in *asf1Δ* (Supplementary Figure 4d). See Page 14.

The old panels e and f in Fig. 6 (which had shown SWI/SNF ChIP in *rtt106Δ*, and Rtt106 ChIP in *snf2Δ*) were now removed since reduction of bindings of SWI/SNF and Rtt106 to the *PDR5* promoter in *rtt106Δ* and *snf2Δ*, respectively, may simply reflect Pdr3 protein level at the *PDR5* promoter rather than cooperative recruitment of SWI/SNF and Rtt106.

Again, I shared the concerns of reviewer 1 over the small shift in nucleosome occupancy over the *PDR5* promoter caused by loss of Snf2. Additionally, Tbp recruitment to *PDR5* was suggested to be impacted by Snf2. The response was to simply re-discuss their use of the data of others. I have no concerns with these other data but this misses the point raised. There may be a slight reduction in binding of either a nucleosome or Tbp but I fail to understand why one would rely on previously published work to validate this essential feature of this current study.

Data mining and further analysis of existing NGS dataset are not just ‘re-discussion’, rather an important aspect of bioinformatics. Since the dataset in question was published from an expert lab in nucleosome positioning, it is sensible to utilise the published dataset in our study. Reviewer 2 has “no concerns with these other data”, so presumably utilising the existing dataset is not an issue. Our analysis of this dataset showed that sliding of TSS-associated nucleosome and reduction in binding of TBP in the absence of Snf2 are statistically significant. A “slight” shift of TSS-associated nucleosome is considered as a significant change in this field (e.g., as shown in Kubik et al. 2019 Nat Struct Mol Biol).

Reviewer 2 says “this misses the point raised”, but we do not understand what is missed here since Reviewer 2 did not explain it. Perhaps it is related to the point raised in the first review from Reviewer 2 “What about the other PDR promoters that don’t respond to Rtt106?”. As we pointed out in our first response, Reviewer 2 misunderstood our data. We analysed the effect of lack of Snf2 (not Rtt106) on nucleosome shift. Nonetheless, we could further assume that Reviewer 2 would like us to compare nucleosome shifts caused by loss of Snf2 on the promoters of PDR genes bound by SWI/SNF with those not bound by SWI/SNF. As shown in the graphs below, we observed a bigger effect of loss of Snf2 on nucleosome shift for the PDR genes bound by SWI/SNF (left panel) than those not bound by SWI/SNF (right panel), as expected. We did not include these figures in our revised manuscript since the yeast genome does not contain enough promoters to draw a statistically significant conclusion, especially for SWI/SNF-unbound PDR genes with strong TBP binding (only 4 gene promoters, data not shown).

I also noted that the ketoconazole induction experiments are all done on media lacking a carbon source while all plate assays contain a carbon source. Nothing was done to address this beyond arguing this is the nutritional condition required to observe induction. These data (induction and drug resistance) are interpreted together yet the growth conditions in which they are observed are quite different. The absence of a carbon source would induce, at a minimum, catabolite derepression and certainly trigger a severe protein degradation phenotype as well as blocks to translation. I do not agree that these vastly different growth conditions may be interpreted together.

As we have already explained in our first response, we had to use the different conditions for testing ketoconazole-induced gene expression and drug resistance. We now modified the main text not to over-interpret gene expression and drug resistance together: on page 12, we now say “Overall, these results suggest that Rtt106 regulates basal expression of *PDR5* via Pdr3 and also mediates drug resistance. Note that it is still unknown whether basal expression of *PDR5* regulated by Rtt106 directly leads to drug resistance and whether Rtt106 also affects drug-induced *PDR5* expression in cells grown long-term with drug.”

We also modified several sentences regarding this issue to avoid over-interpretation, e.g., in the Abstract, from “histone chaperone Rtt106 and the chromatin remodeller SWI/SNF control expression of the PDR network genes, conferring drug resistance.” to “histone chaperone Rtt106 and the chromatin remodeller SWI/SNF control expression of the PDR network genes and confer drug resistance.”.

This manuscript is difficult to integrate into the more than 3 decades of study of the Pdr system as little effort is made to ensure that the current data actually are consistent with the past. Just dismissing these discrepancies as strain differences is not helpful especially when only a single strain is used here. S288c is the strain that was used for genomic sequencing but it is hardly wild-type. It contains a range of mutations including a defective *HAP1* gene. Additionally, work exists using the *S. cerevisiae* knockout disruption mutant strain (BY4742) that is consistent with PDR1 being the key driver of drug resistance (Fardeau, et al (2007) JBC 282: 5063). BY4742 is very closely related to the BY4741 strain used predominantly here. What is normally done when there is some unusual genetics (like Pdr3 being the dominant driver of *PDR5* and drug resistance) as argued here, is to confirm this in other backgrounds. This is not done and isn't addressed since they authors can find one other paper that seems to agree with this.

Reviewer 2 apparently feels strongly about this issue; however Reviewer 2 is also mistaken here because data from Fardeau et al. is actually consistent with our study, and we really don't understand why Reviewer 2 mentioned this paper as conflicting with our study. Fardeau et al. showed that in BY4742 Pdr1 is an important factor for drug-induced expression of the PDR genes (which is completely consistent with our study), but loss of *PDR1* has only a minor effect

on gene expression in the absence of drugs as they say “*PDR1* deletion had few visible effects on gene expression in the absence of drugs” (which is also completely consistent with our study).

To get a consensus whether or not Pdr3 is a dominant driver of *PDR5* expression in the absence of drugs, we carried out further extensive literature search. Of ~133 papers produced by searching for ‘Pdr3’ in PubMed, 21 papers examined *PDR5* expression dependence on either Pdr1 or Pdr3 or both (listed in New Supplementary Table 5). We then found that there is no simple consensus: 6 papers show the dominant driver of *PDR5* expression is Pdr1, while 5 papers show it is Pdr3. It seems that Reviewer 2 is influenced by only some of the papers, e.g., Mahé et al Mol Microbiol 1996, that are frequently cited in review articles. It is clear that there were already inconsistencies about this issue between published papers before our study. We highlight two papers: Mahé et al 1996 from the Kuchler lab and Decottignies et al 1995 JBC from the Goffeau lab, in which they used the exactly same set of strains, but observed opposite results (the former showed that Pdr1 is a dominant driver of *PDR5* expression in the absence of drugs, while the latter showed that Pdr3 is), although both papers were published by expert groups. Also, Katzmann et al 1994 MCB from the Moye-Rowley lab, one of the expert labs, discussed the observed inconsistency of behavior of Pdr1 in the main text: “These large differences in genetic background are a likely explanation for the differences in behavior between the *pdr1* mutant strains”.

The differences in behaviour between Pdr1 and Pdr3 could be due to differences in genetic background, technique used, and/or their combination. Elucidating the reason(s) causing differences seen in published papers is beyond the scope of this study. Our dataset is produced by the most advanced technique, RNA-seq, and is consistent with other published papers with genome-wide analyses (Kemmeren et al. 2014 Cell, Fardeau et al. 2007 JBC). We now explicitly highlight this discrepancy in our paper, which may previously have been missed or misunderstood by some researchers for more than a few decades (page 10-11).

Finally, I found it hard to get around the conclusion that a *pdr1* null prevented azole induced gene expression yet was not as critical as *PDR3* for resistance. Somehow this is explained by Pdr3 being important in basal expression. This seems to be an odd explanation since, at least to me, basal expression is defined as the level of expression of a gene in the absence of some regulating signal (like ketoconazole evidently is for *PDR5*).

We also felt intuitively that those are unexpected data, but as far as we know, there are no published data related to this issue; no published literature comparing azole sensitivity of *S. cerevisiae* strain lacking Pdr1 with that lacking Pdr3 to a wide range of different concentrations of azoles by spot assays. Therefore, what we can do here is to present what we observed. We now modified the text as follows (on page 12). “We further observed that *pdr3Δ* and *rtt106Δ* are sensitive to high doses of ketoconazole, while *pdr1Δ* is sensitive only to low doses but not to high doses (Supplementary Fig. 3f). This finding suggests that Pdr3 and Rtt106 are more critical for resistance to continuous high doses of azole antifungals than Pdr1; and that Pdr1 function alone cannot mediate resistance to high doses at the condition tested. Although these are unexpected results since Pdr1 is critical for drug-induced expression of *PDR5*, this observation might reflect the fact that *PDR3* undergoes auto-activation of its own transcription, while *PDR1* does not.”

These weaknesses led to my lack of enthusiasm for this work.

We now do believe that the issues raised were fully addressed. Our study proposes a new mechanism regulating the PDR gene expression, and we believe our study makes a substantial contribution to this interesting and important field.

REVIEWERS' COMMENTS

Reviewer #2 (Remarks to the Author):

I appreciate the thorough treatment of my comments by the authors and agree that they have have addressed them all.